# UFM: A Simple Path towards Unified Dense Correspondence with Flow

**uniflowmatch.github.io**

**Yuchen Zhang**    **Nikhil Keetha**    **Chenwei Lyu**    **Bhuvan Jhamb**    **Yutian Chen**

**Yuheng Qiu**    **Jay Karhade**    **Shreyas Jha**    **Yaoyu Hu**

**Deva Ramanan**    **Sebastian Scherer**    **Wenshan Wang**

**Carnegie Mellon University**

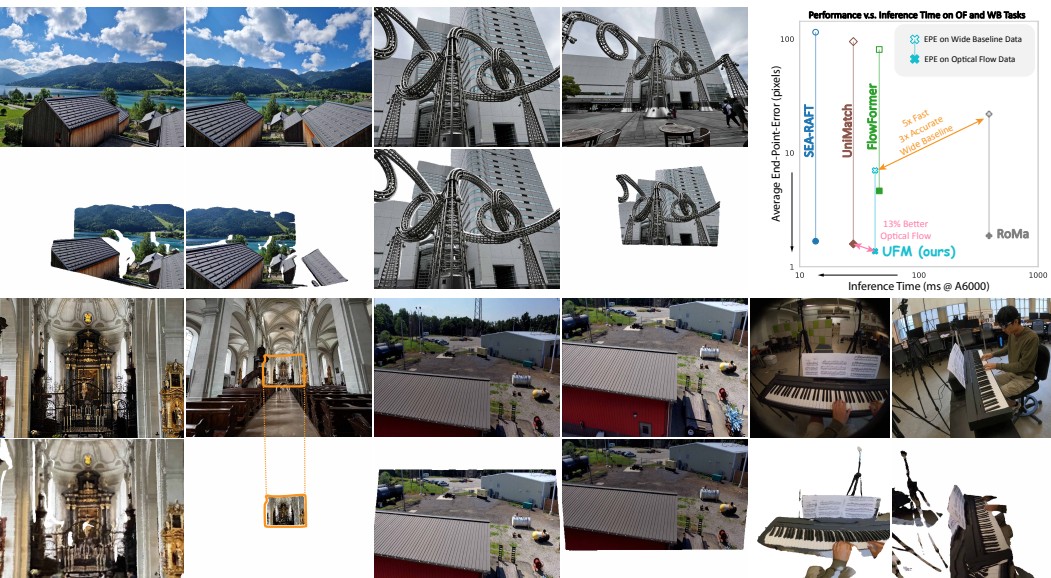

Figure 1: **UFM** (Unified Flow & Matching) unifies dense pixel correspondence tasks such as optical flow and wide-baseline matching. We visualize sets of $2 \times 2$ grids, where the top 2 images are the input, and the bottom 2 are images warped with forward & backward flow. UFM is able to match across a wide range of baselines, including extreme ones with little co-visible overlap.

## Abstract

Dense image correspondence is central to many applications, such as visual odometry, 3D reconstruction, object association, and re-identification. Historically, dense correspondence has been tackled separately for wide-baseline scenarios and optical flow estimation, despite the common goal of matching content between two images. In this paper, we develop a Unified Flow & Matching model (UFM), which is trained on unified data for pixels that are co-visible in both source and target images. UFM uses a simple, generic transformer architecture that directly regresses the $(u, v)$ flow. It is easier to train and more accurate for large flows compared to the typical coarse-to-fine cost volumes in prior work. UFM is 28% more accurate than state-of-the-art flow methods (Unimatch), while also having 62% less error and 6.7x faster than dense wide-baseline matchers (RoMa). UFM is the first to demonstrate that unified training can outperform specialized approaches across both domains. This result enables fast, general-purpose correspondence and opens new directions for multi-modal, long-range, and real-time correspondence tasks.

39th Conference on Neural Information Processing Systems (NeurIPS 2025).

# 1  Introduction

Dense correspondence estimation, which determines where each pixel in one image appears in another, is a core task in computer vision with wide-ranging applications, including visual odometry [45, 47, 58], 3D reconstruction [13, 34, 53], object association [36], place recognition [30, 32, 49], and image warping [71]. Despite its importance, existing methods are typically developed for two separate domains: optical flow, which addresses small displacements between temporally adjacent frames, and wide-baseline matching, which handles large viewpoint or scene changes. This division has led to task-specific models that perform well in one domain but fail to generalize to the other. As a result, these models often break down in real-world scenarios where both small and large motion may co-occur, highlighting the need for unified approaches that bridge this gap.

Existing dense correspondence estimations algorithms have been separated into different tasks. For example, optical flow typically assumes small baselines between the two images, but allows for a dynamic scene, and so often relies on motion priors for temporal consistency. In contrast, wide-baseline matching assumes a static scene but allows for significant changes in viewpoint [37] and time [59], and often require invariant geometric and semantic cues [72]. Despite these differences, both tasks fundamentally aim to establish correspondences between images. This shared objective suggests that they are not inherently separate problems, but rather variations of the same challenge that can be approached within a unified framework.

We are inspired by prior attempts at unifying such correspondence tasks [60, 78], but thus far, none provide a generic solution that outperforms or is on par with specialized solutions. Our experiments suggest that existing work in optical flow and dense wide-baseline matching suffers from biased architectures that are either inefficient when learning from large data or do not have their output format trained/designed for dense, high-resolution output. We aim to answer the question - **can we develop a unified model that benefits from shared training on both optical flow and wide-baseline matching data?** Specifically, what architecture, data, loss, and training scheme do we need to unify flow & matching?

In this work, we scaled a transformer-based regression model over a comprehensive training set of 12 datasets spannning both optical flow and wide-baseline matching. We sample image pairs from our dataset based on covisible content and train exclusively on these regions. We designed a custom geometric sampler with explicit control over viewpoint differences and a filtering pipeline to ensure co-visibility. By restricting supervision to co-visible regions, we discourage the network from relying on global 3D structure alone and encourage correspondence estimation *grounded in visual evidence*. We found that this simple approach leads to a generalizable and efficient model for both optical flow and wide-baseline matching that surpasses most SoTA on its own, achieving further gains with standard refinement techniques.

Finally, to spur further research on correspondence in challenging wide-baseline scenarios, we build a novel dataset for evaluation by holding out environments from the TartanAir-Visual Odometry benchmark [68], using our custom geometric sampler to curate challenging image pairs. Our TartanAir-Wide Baseline (TA-WB) benchmark is a challenging and well-controlled dataset for evaluating dense wide-baseline correspondence.

In summary, our contributions are:

1. For the first time, we demonstrate that unifying the training of both optical flow and wide-baseline estimation can benefits both domains. Our Unified Flow & Matching model (UFM) achieves state-of-the-art performance on benchmarks from both tasks.

2. We find that a generic transformer architecture models unified data better. The simplicity and efficiency of our architecture allows adding existing refinement techniques for further improvement.

3. We introduce a new benchmark, TartanAir Wide-baseline (TA-WB), which evaluates dense correspondence at challenging viewpoint changes.

# 2  Related Works

**Optical Flow Methods**   Optical flow prediction aims to establish dense, pixel-wise motion vectors between temporally adjacent frames. Except for early exploration of optimization-based formulations, current methods are mostly learning-based. Most work has evolved around specialized architectures

including cost volumes [20, 23, 54, 55, 57], coarse-to-fine paradigms [4, 5, 9, 19, 55, 75], and recurrent structures [20–23, 55, 57, 75]. RAFT [57] is one of the most representative works along these ideas. It employs a multi-resolution cost volume between all pairs of patches and a recurrent structure to update the flow prediction iteratively. It has many derivative works [20, 54, 69, 79]. SEA-RAFT [69] is the current state-of-the-art (SoTA) that simplifies RAFT with a regressed initial hypothesis and a multi-modal training objective. Other approaches tried to move beyond these paradigms. FlowFormer [20] uses the transformer architecture to aggregate the cost volume into compact latent tokens for efficient processing. GMFlow [73] casts optical flow into a global matching problem [73, 74, 79] and replaced the costly iterative refinement with a global correlation layer.

In developing a foundation model for generic correspondence prediction, we observed that the specialized architectures of classical optical flow methods struggle with diverse, wide-baseline data, even when trained on it. In contrast, we show that a generic transformer-based regression architecture with sufficient data serves as a robust and generalizable prior. Moreover, it can be effectively combined with these refinement techniques to improve performance further.

**Dense Wide Baseline Methods**    Dense wide-baseline matchers suppress their sparse counterparts since DKM [14], which first obtains a robust, coarse match from patch features and uses regressive warp-refiners to upsample the prediction resolution. RoMa [15] builds upon DKM by using a frozen image foundation model (DINOv2 [46]) for its coarse matching and uses separate convolution-based encoders to provide fine details to warp-refiners. Despite being robust and accurate, both methods have a heavy architecture that limits their application to compute-limited scenarios. We show that our method can achieve similar robustness and accuracy while being about $6\times$ faster.

These methods [14, 15, 41, 61, 62] also include a covisibility mask estimator (some call it "certainty" or "matchability") that helps to exclude matches in occluded or out-of-view regions. This mask is usually directly trained with the ground truth target. We extended this paradigm by computing co-visibility masks for dynamic datasets.

**Unifying Correspondence**    Several work exists in treating correspondence as a unified task. GLUNet [60] is the first work showing that geometric, optical flow, and semantic correspondence tasks can be solved by a unified network. RGM [78] is the most recent work that scaled a RAFT-like architecture on a comprehensive dataset and obtained SoTA zero-shot performance. However, they failed to show that the generalist model, trained on all data, outperforms the specialized model, trained on in-domain data only. Alternative to modeling correspondence densely, COTR [27] took a formulation that predicts one pixel location over each query point, and tested on both optical flow and pose estimation tasks. This formulation is prohibitively expensive for dense flow, and while sparse matches can be interpolated, the resulting performance degrades significantly. In contrast, our work trained a transformer-based architecture that directly regresses dense optical flow and shows mutual benefit between optical flow and wide baseline data.

**Scaling Correspondence**    Recent works have also tried to expand the training dataset for correspondence. Besides the standard optical flow datasets [6, 12, 33, 39, 42, 43], we see a trend in using static wide-baseline matching datasets to pretrain optical flow networks. For example, MatchFlow [11] pretrained on GIM [52], an auto-annotation pipeline that extracts matches from distant frames in real-world videos. Similarly, SEA-RAFT [69] pretrains on TartanAir [68] and observed improved generalization. Existing work in wide-baseline matching [18, 28, 64] has also expanded the dataset towards more modalities such as satellite, IR, depth, event, and medical. Although they have shown successful matching between challenging modalities, they do not show that scaling with additional data helps improve the original RGB-RGB matching.

Recent advancements in end-to-end learning have also encouraged scaling a generic architecture for 3D reconstruction and correspondence [26, 29, 31, 45, 65, 67]. Perceiver IO [24] shows its architecture can solve diverse vision tasks, including optical flow. CroCoV2 [70] also shows that optical flow can be directly regressed from its backbone pre-trained on the cross-image-completion task. However, CrocoV2 stopped at low resolution and both methods required a sliding window method to infer at high resolution, which failed to capture correspondences across windows. Furthermore, CroCoV2 doesn't train the two-view transformer from scratch to directly regress flow. More recent follow-up MASt3R [34] finetuned DUSt3R [67] to output pixel-wise feature descriptors and proposed a fast reciprocal matching to decode sparse matches efficiently. However, this paradigm does not provide dense matches and is prohibitively slow without subsampling.

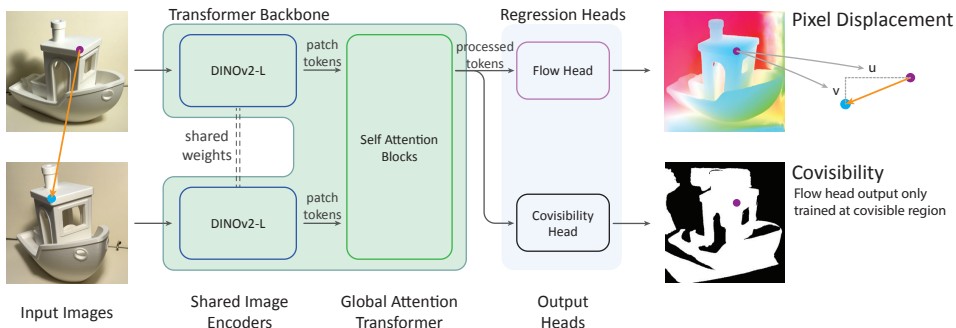

Figure 2: **The UFM Architecture**: Two images are encoded by a shared DINOv2 encoder into patch features, concatenated, and then processed by 12 self-attention transformer layers. Intermediate tokens are decoded by separate DPT heads to regress pixel displacement and covisibility maps, representing correspondence and visibility across views.

## 3 Unified Flow & Matching Model

**3.1. UFM Architecture.** Given two images $I_1, I_2 \in \mathbb{R}^{3 \times H \times W}$ as input, our Unified Flow and Matching (UFM) model (Fig. 2) predicts the visually grounded dense correspondences and covisibility:

$$\{\phi_1, C_1\} = f_{UFM}(I_1, I_2) \tag{1}$$

where $\phi_1 \in \mathbb{R}^{2 \times H \times W}$ is a forward pixel displacement map (flow) which maps each $[u, v]$ position in $I_1$ to a continuous position in $I_2$ and $C_1 \in \mathbb{R}^{1 \times H \times W}$ is a binary mask, where each value indicates if the $[u, v]$ position in $I_1$ is visible in $I_2$.

In particular, UFM employs a simple end-to-end transformer with a CNN head to directly regress the flow value, unlike prior methods [15, 69, 74] that first construct an initial solution at the patch level and then progressively upsample to the pixel level. We empirically find the paradigm used in prior methods to yield suboptimal accuracy, as shown in Appendix F. Beyond accuracy, direct flow regression offers additional advantages: (1) Performing attention at the patch level is fast for flow prediction, while the DPT [48] enables predictions directly at the pixel level. (2) Structural simplicity enables easy optimization and potential for additional simple pixel-level refinements without a huge impact on efficiency. We elaborate on the end-to-end transformer further below.

**Feature Encoding:** Amongst various image encoders, we find DINOv2 ViT-L [46] to be the most optimal. DINOv2 takes as input images and predicts patch tokens $F_E \in \mathbb{R}^{1024 \times H/14 \times W/14}$. Given the two sets of patch tokens, we fuse them with a view index positional encoding unique to each view and then apply 12 successive layers of self-attention. While other prior methods [34, 67, 70] employ cross-attention blocks, which in theory have the same compute requirement as our design, we find that the self-attention transformer is better accelerated by Flash-Attention [8] due to its longer sequence length. This leads to better training and inference efficiency. Also, we empirically find that both types of transformers have similar performance in terms of flow regression.

**Predicting Flow & Covisibility:** After the self-attention transformer is applied, we employ two separate DPT heads which take as input the encoded patch tokens from $I_1$ and respectively predict the flow $\phi_1$ and logits for the covisibility mask $C_1^{logits}$. We empirically find that employing a single DPT head for both flow and covisibility prediction leads to degraded performance. The DPT inputs the output features from the DINOv2 image encoder and the self-attention transformer's 6th, 9th, and 12th layer features. The final predicted covisibility is obtained by $C_1 = sigmoid(C_1^{logits})$.

**Refinement by Classification:** While we find that the regression of dense correspondence (flow) is robust, it is not always precise (e.g., see average EPE & outlier numbers for UFM 560 in Table 2). Hence, we designed a simple classification-based local refinement technique to improve the accuracy of UFM's inlier predictions. We take inspiration from MASt3R [34]'s design to regress pixel-wise matching features based on transformer backbone features. Additionally, to capture fine details for the refinement, we added a U-Net encoder following RoMa [15]. As shown in Fig. 3, we differ from

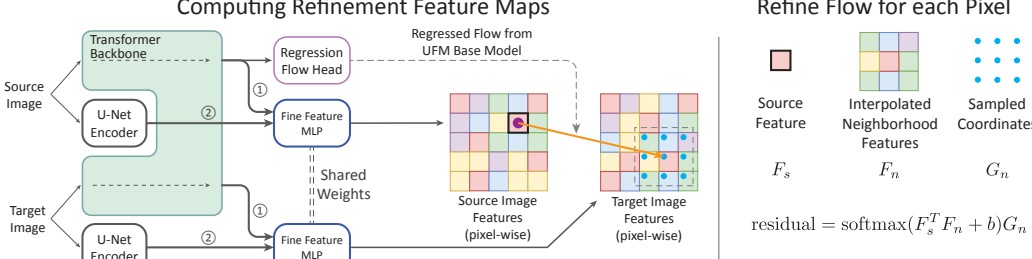

Figure 3: **Refinement of Correspondence by Classification:** We compute a per-pixel feature map by combining (1) globally aligned features from the UFM backbone and (2) local fine features encoded by a separate U-Net. For each pixel in the source image, we first use the regression flow target to interpolate features around a local neighborhood. We then compute the attention between the source features and the features from the local neighborhood, and use it to weight-add the coordinates as a refinement value. $b$ is a constant attention bias.

MASt3R [34] in how we leverage the refinement features for correspondence: as opposed to matching dense features across the entire image (global search), we use the regressed flow from UFM's DPT to guide the feature matching around a small $7 \times 7$ neighborhood, thereby leading to $60\times$ efficiency over MASt3R (Table 2), or about $24\times$ considering the follow-up acceleration Speedy MASt3R [35]. In particular, we compute the attention between each pixel and its local $7 \times 7$ neighborhood determined by the regressed flow and use the weighted sum of coordinates by the softmax attention as the residual to update the initially regressed flow.

**3.2. Training Objective.** To train UFM, we supervise the predicted pixel displacement map $\phi_1$ and the covisibility mask $C_1$. Importantly, supervision of the correspondence is restricted to covisible pixels. This design encourages the model to ground correspondence in visual evidence, rather than inferring 3D geometry from a single view and extrapolating into occluded or out-of-view regions.

We trained with a robust regression loss [1], following the approach in RoMa [15], which focuses its gradient on inlier predictions with small errors—typically around 1 to 2 pixels. We selected this loss for two main reasons. First, it encourages precise learning from reliable matches by emphasizing small residuals. Second, it reduces the impact of incorrect data during training, as robust losses exhibit vanishing gradients for large flow errors, which are commonly caused by unmatchable pairs. Specifically, we used the generalized Charbonnier loss with parameters $\alpha = 0.5$ and $c = 0.24$. See Appendix H for more visualization and discussion.

$$L_{\text{EPE}}(\phi_1, \phi_1^{gt}) = \frac{1}{\sum_{i \in I} C[i]^{gt}} \sum_{i \in I} C[i]^{gt} l_{\text{robust}}(\|\phi_1 - \phi_1^{gt}\|_2) \tag{2}$$

As we supervise the network only on covisible pixels, the network can have an arbitrary output in the non-covisble pixels during usage. Hence, we also predict a covisible mask to exclude outputs from these regions during usage. To train this mask, we used the standard binary cross-entropy loss.

$$L_{\text{BCE}} = \frac{1}{H * W} \sum_{i \in I} \left[ -C[i]^{gt} \log(C_1^{logits}) - (1 - C[i]^{gt}) \log(1 - C_1^{logits}) \right] \tag{3}$$

We find that upweighting the covisibility loss by a factor of 10 is optimal for the prediction of good covisibility and doesn't impact flow estimation quality. Thus, our final loss is $L = L_{\text{EPE}} + 10 \times L_{\text{BCE}}$.

**3.3. Combining Flow and Matching Datasets.** We compiled a unified dataset consisting of 12 datasets spanning diverse sources, motion patterns, and environments from both wide-baseline matching and optical flow domains, as detailed in Tab. 1. The collection of these datasets features diverse indoor, outdoor, in-the-wild, and dynamic scenes.

Each dataset was carefully vetted for depth consistency and geometric correctness, as not all are suitable for precise training and evaluation. For example, we found that ARKitScenes [2] contains

Table 1: **Diverse suite of dense correspondence datasets used to train UFM.**

| Dataset | Images | Pairs | Scenes | Source | Dynamic | Wide Baseline | Frame to Frame | Pairs in Epoch |
|---|---|---|---|---|---|---|---|---|
| BlendedMVS [76] | 115 k | 1.15 M | 503 | Mesh Reconstruction | ✗ | ✓ | ✗ | 100 k |
| MegaDepth [37] | 38.8 k | 1.8 M | 275 | COLMAP MVS | ✗ | ✓ | ✗ | 100 k |
| TartanAirV2 [68] | 1.37 M | 688 k | 55 | Synthetic | ✗ | ✓ | ✗ | 100 k |
| Scannet++ V2 [77] | 265 k | 14.3 M | 295 | Laser Scan | ✗ | ✓ | ✗ | 100 k |
| Habitat CAD [56] | 201 k | 175 k | 91 | CAD Reconstruction | ✗ | ✓ | ✗ | 25 k |
| StaticThings [51] | 22.4 k | 337 k | 2250 | Synthetic | ✗ | ✓ | ✓ | 10 k |
| Kubric4d [17, 63] | 2.4 M | 9 M | 2800 | Synthetic | ✓ | ✓ | ✓ | 50 k |
| FlyingThings [39] | 22.4 k | 20.2 k | 2239 | Synthetic | ✓ | ✗ | ✓ | 50 k |
| FlyingChairs [12] | 44.4 k | 22.2 k | 964 | Synthetic | ✓ | ✗ | ✓ | 25 k |
| Spring [40] | 10 k | 9.9 k | 30 | Synthetic | ✓ | ✗ | ✓ | 25 k |
| Monkaa [39] | 8.6 k | 8.6 k | 24 | Synthetic | ✓ | ✗ | ✓ | 5 k |
| HD1K [42] | 1081 | 1046 | 35 | Real | ✓ | ✗ | ✓ | 5 k |

inconsistent depth estimates, leading to flow errors of up to 5 pixels, which is unacceptable in the matching domain where methods aim for sub-pixel accuracy.

In general, we paired the data for a well-distributed range of covisibility and optical center difference. For most of the static wide-baseline datasets, we followed the pairing scheme in DUSt3R [67] and CUT3R [66] and used adjacent frames for optical flow datasets. We selected the ratio from each dataset largely based on the number and quality of the scenes. We further provide details on sampling pairs for ScanNet++ V2 [77] & Kubric4D [17] in the supplementary. Notably, we mined new pairs from Kubric4D across both time and viewpoint, making it the only dataset in our collection that is both dynamic and wide-baseline.

Because we aim to develop a unified model that generalizes concurrently to both optical flow and wide-baseline matching domains, we train on both types of data simultaneously. This allows examples from both domains to appear within a single gradient update, promoting cross-domain generalization. We computed covisibility and correspondence for all pairs of images to support this unified training.

We compute correspondence targets and covisibility mask differently depending on the dataset type, accounting for the specific characteristics of posed image collections and optical flow labels in static scene data and synthetic data such as Kubric. This process is detailed in the supplementary.

**TA-WB Training & Benchmarking Dataset:** We developed a special geometric sampler for the TartanAirV2 [68] dataset to sample geometrically challenging yet covisible pairs (further described and samples provided in the supplementary). Since TartanAirV2 provides images covering all six sides around each camera center, all visual information is preserved, and we can resample virtual cameras with arbitrary orientations. Our sampler utilizes this freedom to control the viewpoint difference explicitly. We check all sampled pairs for matchability and reject occluded or textureless pairs (for e.g., two cameras facing white walls). We made the final samples to equally distribute the camera optical center angle difference between $0°$ and $120°$.

**3.4. Training Details.** We train the network with a longest side resolution of 560 (with aspect ratios varying from 3:1 to 1:1) for 48 epochs with data as specified in Table 1. All our datasets permit academic research, and the publicly released UFM model weights will be licensed following this. We use a peak learning rate of $1 \cdot 10^{-4}$ for the global attention transformer and DPT heads and $5 \cdot 10^{-6}$ for the encoder to preserve DINOv2 pre-training. This contrasts with the frozen DINOv2 used by RoMA [15] (which we find suboptimal), and we provide further insights in the supplementary. We use AdamW optimizer with a cosine decay learning rate schedule using 10% linear warmup, 0.05 weight decay, and $\beta = \{0.9, 0.95\}$. Since most of our data has bidirectional correspondence, we symmetrize the batches. This leads to an effective batch size of 96 pairs, where half of them are unique. The training takes 4 days on 8 H100 GPUs. We name this checkpoint as UFM 560.

Some downstream tasks, like visual odometry [47], require sub-pixel accuracy, making high-resolution images essential. However, training at high resolution is computationally expensive. To address this, we bootstrap a high-resolution model, UFM 980, from UFM 560. The wide-baseline datasets do not have depth annotations at a high resolution (1K), and upsampling the pre-computed flow at lower resolutions would be sub-optimal for sub-pixel training. Hence, we train with $10\times$ lower learning rates than the 560 training on all optical flow data for 15 epochs. Furthermore, we change the supervision range to all pixels to follow the standard evaluation protocol in optical flow.

Table 2: **Wide Baseline Dense Correspondence:** Zero-shot dense correspondence evaluation at all covisible pixels. We report the AEPE and outlier rates at thresholds of 1, 2, and 5 pixels. UFM outperforms all dense methods by a large margin and matches MASt3R's performance, despite MASt3R's advantage in selecting its confident pixels, while being 60× faster (24× considering follow up [35]).

| Method | Eval Range | ETH3D | | | | DTU | | | | TA-WB | | | | Runtime |
|---|---|---|---|---|---|---|---|---|---|---|---|---|---|---|
| | | EPE ↓ | 1 px ↓ | 2 px ↓ | 5 px ↓ | EPE ↓ | 1 px ↓ | 2 px ↓ | 5 px ↓ | EPE ↓ | 1 px ↓ | 2 px ↓ | 5 px ↓ | ms ↓ |
| SEA-RAFT | | 113.13 | 80.4 | 71.8 | 63.6 | 58.91 | 72.4 | 60.4 | 50.3 | 172.12 | 90.0 | 84.6 | 80.1 | 13.6 |
| FlowFormer | | 74.83 | 80.4 | 69.1 | 58.4 | 41.14 | 77.1 | 62.2 | 47.4 | 126.65 | 88.0 | 78.8 | 70.8 | 46.5 |
| UniMatch | Covisible | 91.21 | 73.1 | 64.5 | 56.7 | 48.98 | 69.2 | 57.0 | 46.9 | 144.54 | 87.2 | 80.5 | 75.0 | 28.2 |
| RoMa | Pixels | 7.94 | 51.1 | 33.4 | 19.9 | 9.69 | **52.1** | 33.8 | 19.9 | 48.10 | 63.7 | 47.7 | 39.8 | 387.4 |
| UFM 560 | | 2.64 | 46.5 | 23.9 | **8.7** | 5.56 | 58.4 | 33.6 | **13.2** | 12.87 | 53.5 | 31.8 | **17.0** | 42.9 |
| UFM 560 - refine | | **2.60** | **44.2** | **22.8** | **8.7** | **5.55** | 55.5 | **32.9** | 13.8 | **12.84** | 51.4 | 30.6 | **17.0** | 70.1 |
| MASt3R | MASt3R's | 1.31 | 33.4 | 11.6 | **2.0** | 2.23 | 50.1 | **20.6** | **5.3** | 6.21 | 54.8 | 22.5 | **6.2** | 2517.8 |
| UFM 560 | Output | 1.34 | 31.7 | 12.1 | 3.1 | 2.30 | 49.2 | 23.5 | 6.3 | 6.19 | 42.1 | 19.5 | 7.4 | 41.0 |
| UFM 560 - refine | | **1.29** | **29.0** | **11.1** | 3.1 | **2.18** | **42.6** | 20.8 | 6.2 | **6.13** | **38.7** | **17.8** | 7.4 | 56.1 |

Table 3: **Relative Pose Estimation:** Area Under the Curve results for pose estimation on zero-shot datasets (ETH3D, Scannet 1500) and our proposed benchmark TA-WB (zero-shot scene assets, appearance & geometry). Gray text indicates results where the evaluation dataset is in the training set.

| Method | ETH3D | | | Scannet-1500 | | | TA-WB | | |
|---|---|---|---|---|---|---|---|---|---|
| | AUC @ 5° ↑ | @ 10° ↑ | @ 15° ↑ | AUC @ 5° ↑ | @ 10° ↑ | @ 15° ↑ | AUC @ 10° ↑ | @ 20° ↑ | @ 30° ↑ |
| RoMa | 63.7 | 74.2 | 78.6 | 29.2 | 50.0 | 60.9 | 2.2 | 11.4 | 23.2 |
| MASt3R | 65.7 | 77.0 | 81.5 | 34.2 | 57.2 | 68.0 | **2.5** | 13.3 | 27.9 |
| UFM 560 | 61.6 | 74.1 | 79.3 | 30.7 | 53.5 | 64.8 | 2.3 | 13.3 | **28.6** |
| UFM 560 - refine | **66.7** | **77.1** | **81.6** | 31.6 | 54.1 | 65.3 | **2.5** | **13.5** | **28.6** |

# 4   Benchmarking Unified Dense Correspondence

**4.1. Zero-Shot Wide-Baseline Correspondence.**   We perform direct evaluation via dense correspondences and indirect evaluation via pose estimation. We compare all covisible correspondence to the ground truth and report Average End-Point-Error (AEPE) and outlier rates. We use exhaustively sampled covisible pairs from ETH3D [50], DTU [25], and TA-WB. For pose estimation, we evaluated on ETH3D, TA-WB, and Scannet-1500 [7]. While pose estimation benchmarking is the standard practice, we believe that dense wide-baseline EPE provides a more direct and stable measure of matching quality by eliminating the influence of confidence prediction and sampling.

**Baselines**   We benchmark against SoTA, including RoMa [15] (indoor, for better performance) and MASt3R [34]. MASt3R is a sparse method that only provides correspondence passing its cycle-consistency check. We adjusted its subsampling to get the most dense output and evaluated UFM on the same set of reported pixels. While this setup favors MASt3R by restricting evaluation to it's confident matches, it enables comparison with one of the most robust sparse matches. We include optical flow methods for completeness, comparing against SEA-RAFT [69], FlowFormer [20], and GMFlow [73], using their checkpoints trained on all available data.

**Dense Wide Baseline Results**   In Table 2, despite giving MASt3R an advantage, we showcase that UFM significantly outperforms all dense methods in precision, while achieving nearly 60× lower runtime (24× considering follow up [35]) than MASt3R — the only method with comparable precision. Furthermore, UFM significantly outperforms all dense methods, achieving on average 62% less EPE and 6.7× better runtime compared to the best dense baseline, RoMa.

**Pose Estimation Results**   We follow DKM [14] and evaluate UFM for pose estimation. As shown in Table 3, UFM achieves the best accuracy on ETH3D and TA-WB benchmark, and second place on Scannet-1500 (despite not being trained on this dataset). This performance shows that UFM's correspondence is well-balanced and suitable for 3D geometric tasks. Moreover, we observe a notable improvement by adding refinement on top of UFM, highlighting the potential for integrating other refinement techniques on top of the base model for further improvement.

**4.2. Optical Flow Correspondence.**   We evaluate zero-shot optical flow performance on Sintel and KITTI-2015 training set. We evaluate on both covisible pixels and all pixels which is the standard

Table 4: **Optical Flow Estimation:** Zero-shot evaluation across covisible ([covis]) and all pixels ([all]) on the Sintel and KITTI training sets. Each method is inferred at different resolutions, and the metrics are computed at the dataset's original resolution (1K) and on an A6000 Ada GPU.

| Method | Inference Resolution | Sintel Clean | | | | | Sintel Final | | | | | KITTI | | | Runtime |
|---|---|---|---|---|---|---|---|---|---|---|---|---|---|---|---|
| | | EPE ↓ [covis] | EPE ↓ [all] | 1px ↓ | 3px | 5px | EPE ↓ [covis] | EPE ↓ [all] | 1px ↓ | 3px | 5px | F1 EPE ↓ [covis] | F1 EPE ↓ [all] | F1 % ↓ | ms ↓ |
| SEA-RAFT | | 0.49 | 1.27 | **7.4** | **3.4** | 2.5 | 2.28 | 3.86 | **13.1** | 7.7 | 6.1 | 2.10 | 4.29 | 14.3 | 20.7 |
| FlowFormer | | 0.47 | 1.01 | 8.7 | 3.6 | 2.5 | 1.43 | 2.38 | 14.0 | 7.4 | 5.5 | 3.75 | 6.03 | 15.8 | 155.1 |
| Unimatch | 1K | **0.43** | **0.96** | **7.4** | **3.4** | **2.4** | 1.63 | 2.70 | 13.4 | 7.4 | 5.6 | 2.38 | 4.92 | 17.5 | 76.7 |
| UFM 980 | | 0.61 | 1.16 | 11.7 | 4.5 | 3.0 | 1.28 | 2.04 | 14.9 | **7.1** | **5.1** | **2.05** | **2.94** | **11.0** | 122.9 |
| UFM 980 - refine | | 0.56 | 1.15 | 10.2 | 4.6 | 3.3 | **1.25** | **2.01** | 15.0 | 7.2 | **5.1** | **2.05** | 2.96 | **11.0** | 213.9 |
| SEA-RAFT | | 0.65 | 1.47 | 10.5 | 4.5 | 3.2 | 2.24 | 3.69 | 15.5 | 8.5 | 6.6 | 2.36 | 4.21 | 15.5 | 14.7 |
| FlowFormer | | 1.88 | 2.92 | 23.6 | 10.1 | 7.2 | 7.39 | 8.92 | 35.1 | 21.5 | 17.5 | 4.64 | 7.89 | 29.3 | 77.5 |
| Unimatch | 560 | **0.60** | 1.20 | 10.3 | 4.2 | 2.9 | 1.73 | 2.76 | 16.0 | 8.0 | 5.9 | 2.43 | 4.66 | 17.7 | 30.0 |
| RoMa | | 1.18 | | | | | 2.13 | | | | | 2.30 | | | 390.3 |
| UFM 560 | | 0.79 | | Trained on covisible pixels only | | | 1.44 | | Trained on covisible pixels only | | | 1.87 | | Trained on covisible pixels only | 44.0 |
| UFM 560 - refine | | 0.72 | | | | | **1.40** | | | | | **1.69** | | | 57.0 |

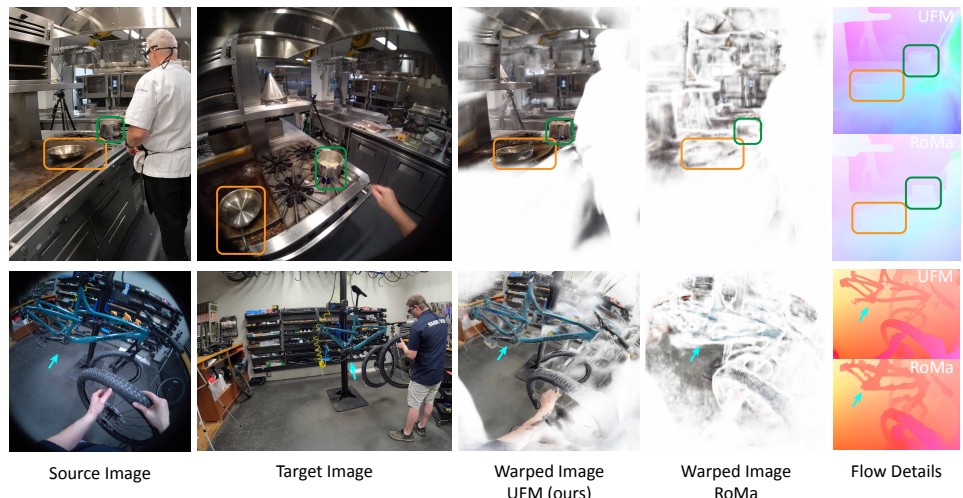

Source Image | Target Image | Warped Image UFM (ours) | Warped Image RoMa | Flow Details

Figure 4: **UFM on Ego-Exo 4D** [16]: UFM succeeds in matching out-of-distribution environments, camera models, and challenging viewpoint shifts, showcasing its strong generalization.

protocol that includes occluded and out-of-bound pixels. On Sintel, we report the AEPE for both cases and the ratio of pixels with EPE above 1, 3, and 5 pixels for all pixels.

**Baselines** We compare our approach to all optical flow methods in Section 4.1 and RoMa, using the checkpoint trained on FlyingChairs, FlyingThings, and TartanAir (SEA-RAFT only)—i.e., the best trained model before violating the zero-shot setting.

**Results** Table 4 shows that UFM, *without any refinement*, achieves state-of-the-art zero-shot performance on Sintel-Final and KITTI in terms of both EPE and most pixel outlier metrics, while also delivering competitive performance on Sintel-Clean. These results demonstrate that the UFM base model has strong generalization and precision to be combined with existing refinement techniques.

**4.3. Generalizable Matching on Ego-Exo 4D.** We ran UFM on images from the Ego-Exo4D [16], which features videos captured in first and third person view across diverse scenes. As shown in Fig. 4, compared to RoMA, UFM achieves strong generalization & robust matching.

**4.4. Insights towards Unified Correspondence.**

**Data:** We conduct an ablation study to see if UFM benefits from unified training on merged data as opposed to training on specialized data only. Specifically, we train UFM only on optical, wide baseline, and the combination for $100 + 20$ epochs at 224 & 560 resolutions. Across the different variants, UFM sees each data point the same number of times. Although the total number of gradient steps differs, the number of epochs is large enough for the training to have effectively converged. We then evaluate optical flow and dense wide-baseline performance as in the previous sections.

Table 5: **Unified optical flow (OF) and wide-baseline (WB) training leads to mutual improvement.**

| Pretrain Data | Optical Flow Tasks | | | Wide Baseline Tasks | | | | | |
| | Sintel-C EPE | Sintel-F EPE | KITTI EPE | DTU 1px | DTU 2px | ETH3D 1px | ETH3D 2px | TA-WB 1px | TA-WB 2px |
| --- | --- | --- | --- | --- | --- | --- | --- | --- | --- |
| OF | 1.27 | 1.81 | 15.57 | 91.4 | 80.4 | 96.4 | 91.6 | 98.4 | 94.9 |
| WB | 1.66 | 2.24 | 3.13 | 70.5 | 42.8 | 54.5 | 28.4 | 61.5 | 35.3 |
| OF + WB | **1.02** | **1.48** | **2.62** | **69.0** | **41.4** | **52.4** | **27.0** | **59.2** | **34.2** |

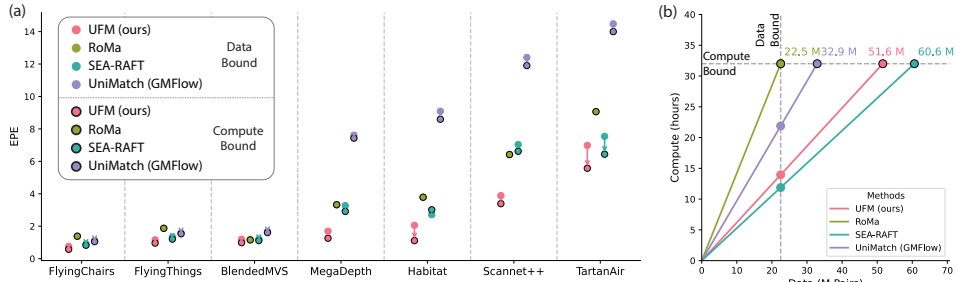

Figure 5: **Architecture Ablation:** Validation EPE for various architectures trained on the same $224 \times 224$ resolution data as UFM. We report performance on different val sets at Data Bound (22.5 M pairs) or Compute Bound (at 32 hours on 8 H100 GPU) **(a) Validation Set Performance**: When trained on more difficult data (such as TartanAir), UFM significantly outperforms alternatives for both bounded data and compute. **(b) Training Speed Comparison**: We plot the number of pairs seen during training as a function of compute, and label the number of pairs that each architecture can train on at compute bound. UFM is far more efficient than most methods (except SEA-RAFT).

In Table 5, UFM outperforms it's own specialized variants, thereby indicating a **mutual improvement** when merging the two data types. For optical flow, we observe that adding wide-baseline data brings a $20\% - 80\%$ decrease in EPE, especially on the KITTI dataset. For wide-baseline, we observe that adding optical flow data brings a 3.2% relative decrease in 1, 2 pixel outlier rates.

**Architecture:** To test the scalability of existing architectures, we trained SEA-RAFT, UniMatch, RoMa, and UFM on the same unified data (Table 1). Each architecture is trained with its original loss functions as specified in the respective papers. We recorded the validation set EPE at data bound (35 epochs, 22.5 M pairs) and compute bound (32 hours on 8 H100 GPU) to measure scalability. Fig. 5 shows that UFM performs best on all datasets at both data and compute bound. This indicates the benefits of using a simple architecture to scale on large amounts of data, where UFM shows significantly increasing performance with compute on harder datasets like ScanNet++ and TartanAir. We believe UFM's effectiveness in scaling to the unified dataset is crucial for achieving mutual benefit across opticalflow and wide-baseline matching.

## 5  Limitations

While UFM represents an exciting development in constructing models for unified dense image correspondence, some limitations remain with semantic matching capabilities. As shown in Fig. 6, on WxBS [44], we find that UFM works on challenging image pairs that demonstrate scale, viewpoint, texture, and illumination and tends to struggle with extreme seasonal changes and matching across spectrums, i.e., visual to very dark infrared (thermal). RoMA [15] is robust to such semantic changes due to the coarse patch correlation provided by *frozen* DINOv2 [46] features, with the help of additional fine features from ConvNet in its upsampling process. We find that freezing the encoder does not benefit an end-to-end transformer architecture such as UFM. As shown in Appendix E, we find that freezing the pre-trained DINOv2 image encoder leads to a significant drop in dense correspondence performance. Opportunities may lie in complementing frozen DINOv2 features from other sources. Furthermore, although we constrain the learning rate to remain relatively small to preserve DINOv2's pretraining, we find that DINOv2 can still deviate significantly during extensive training and lose some of its semantic matching abilities. Through preliminary exploration, we find that this can be mitigated by incorporating semantic matching data, semantic preservation losses, or

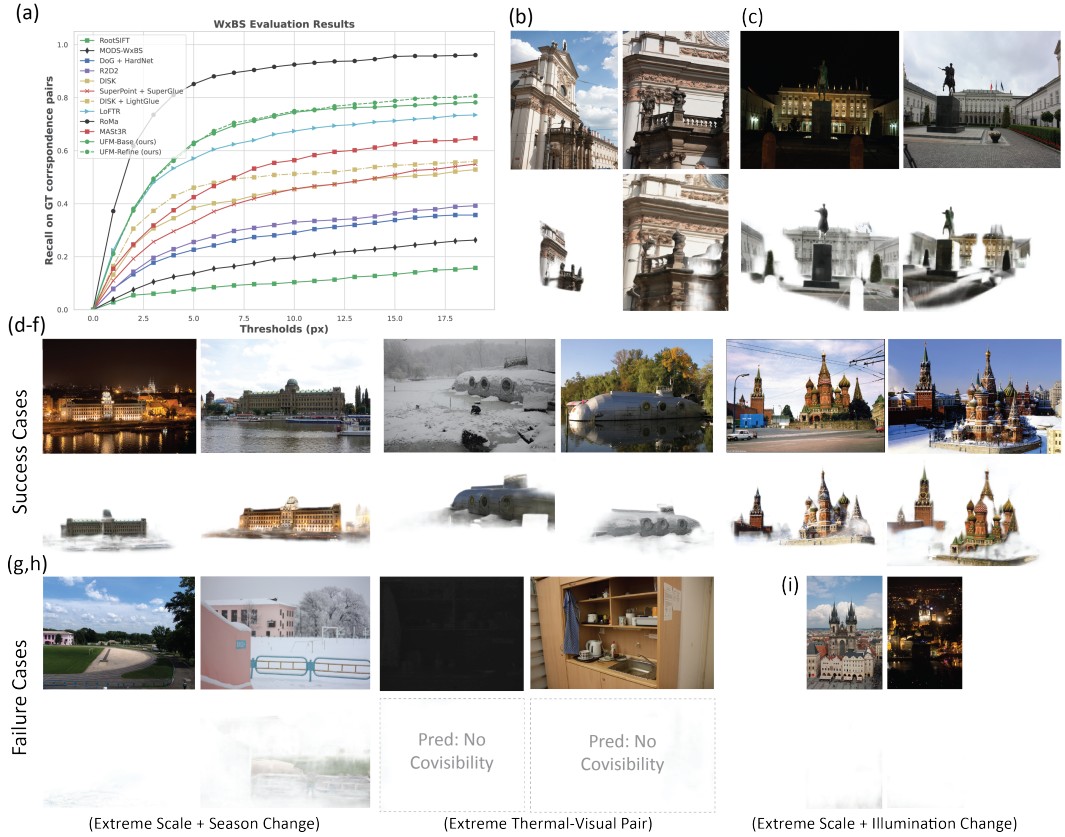

Figure 6: **WxBS Benchmarking [44]:** We find that UFM: (a) outperforms MASt3R [34], another end-to-end transformer trained on large-scale data for correspondence; (b-f) performs well on images with scale, viewpoint, illumination, and seasonal changes, and (g-i) struggles with pairs showing extreme coupled season, illumination, and scale changes or captured across different imaging spectrums, where RoMA [15] is more robust. We provide further insights in Section 5 and believe the primary reason to be the preservation of semantic matching capabilities in the pre-trained image encoder.

specialized fine-tuning that limits the extent of deviation from the pre-trained weights. We aim to address this in future releases of UFM.

We also acknowledge that although our training set and the TA-WB evaluation is geometrically challenging, they are biased toward static objects, potentially reducing the accuracy on dynamic ones, as shown in Appendix G. Future work may include more sceneflow datasets, such as ParallelDomain-4D [63], to balance the distribution.

## 6 Conclusion

We present UFM, a Unified Flow and Matching model that predicts visually grounded dense correspondences and covisibility. Using a simple transformer-based design, UFM directly regresses high-resolution correspondence and covisibility maps, enabling it to learn from a unified dataset effectively. Extensive Experiments show that UFM, trained on optical flow and wide-baseline matching data, benefits from mutual improvement and outperforms specialized methods in each domain. Looking ahead, combining UFM with semantic matching and refinement techniques would further improve its robustness and accuracy, paving the way to general-purpose correspondence prediction.

## Acknowledgments

This work was supported by Defense Science and Technology Agency (DSTA) Contract #DST000EC124000205 and partially by DEVCOM Army Research Laboratory (ARL) under SARA Degraded SLAM CRA W911NF-20-S-0005. The compute for this work was provided by Bridges-2 at PSC through allocation cis220039p from the Advanced Cyberinfrastructure Coordination Ecosystem: Services & Support (ACCESS) program, which is supported by NSF grants #2138259, #2138286, #2138307, #2137603, and #213296. We thank Swaminathan Gurumurthy, Mihir Sharma, Jeff Tan, Shibo Zhao, Can Xu, Khiem Vuong, and other members of the AirLab for their insightful discussions and assistance with parts of the work. Lastly, shout out to Peter Kontschieder for one of the in-the-wild image pairs featured in the first figure.

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

Table S.1: Underlying data sources used for **generating correspondence and covisibility ground truth**, along with the reprojection error threshold used when using depth and pose for covisibility.

| Category | Dataset | Source of Correspondence | Source of Covisible Mask | Abs. Depth Threshold $\tau_d$ | Rel. Depth Threshold $\tau_r$ |
|---|---|---|---|---|---|
| *Static Scene* | BlendedMVS [76] | | | 0.1 | 0.005 |
| | MegaDepth [37] | | | 0.1 | 0.005 |
| | TartanAir V2 [68] | Unproject depthmap across cameras | Threshold depth reprojection error | 0.1 | 0.01 |
| | ScanNet++ V2 [77] | | | 0.1 | 0.005 |
| | Habitat CAD [56] | | | 0.1 | 0.005 |
| *Optical Flow* | Spring [40] HD1K [42] | Dataset-provided | Dataset-provided | | |
| | FlyingThings [39] | Dataset-provided | Scene flow + reprojection threshold | 0.01 | 0.001 |
| | Monkaa [39] | | | 0.01 | 0.001 |
| | FlyingChairs [12] | Dataset-provided | FoV mask (approximate) | | |
| *Rigid Posed Objects* | Kubric4D [17, 63] | Depthmap & object pose | Depthmap & object pose + reprojection threshold | 0.1 | 0.005 |

## A    Computing Covisibility Mask

Computing the covisibility mask for all datasets in Table 1 is a key step to support unified training. In this section, we detail the exact protocol and parameters we used to compute the covisibility mask for all datasets, summarized in Table S.1. We will begin with a general principle of using depth reprojection error to compute covisibility, and then detail its application to three data categories: (1) Static Scenes, (2) Optical Flow, and (3) Rigid Posed Objects.

**A.1. Covisibility from Depth Reprojection Error.**    Given two corresponding pixels in the source and target images, we determine their covisibility by checking 3D consistency - that is, whether their depths unproject to the same 3D point. We compute the Euclidean distance between the points, and consider the pixels covisible if their distance is below a threshold. We refer to this approach as thresholding depth reprojection error.

Formally, given a source pixel $i_s \in I_1$ and a target pixel $i_t \in I_2$, we compute their 3D coordinates $p_s$, $p_t$ and the depth reprojection error $e$, defined as $e = \|p_s - p_t\|_2$. Then, the pixels are determined to be covisible if $\|p_s - p_t\|_2$ is less than an absolute threshold $\tau_d$.

While this metric captures the fundamental idea of computing 3D consistency, it implies a fixed 3D tolerance regardless of the scene distance from the camera. We found this is suboptimal when handling both near and far objects, as far objects are described with less pixels, thus having larger uncertainty in depth and geometry. To address this, we introduce a relative threshold that increases linearly with the distance between the source 3D point $p_s$ and the target camera center $O_2$. Thus, the final thresholding scheme we use is:

$$e = \|p_s - p_t\|_2 < \tau_d + \tau_r \cdot \|p_s - O_2\|_2 \tag{S.1}$$

All dataset categories use the same covisibility thresholding scheme, with dataset-specific parameters summarized in Table S.1. They only differ in how the 3D points $p_s$ and $p_t$, and ultimately the error $e$, are computed. We describe these procedures in the following subsections.

**A.2. Computing Correspondence and Covisibility.**    We begin by specifying the relevant information required from each dataset category, followed by an explanation of how the corresponding 2D pixel $i_t$, the 3D points $p_s$ and $p_t$, and the reprojection error $e$ are computed given a source pixel $i_s$.

**Static Scenes**    For static scenes, the fixed geometry allows us to compute covisibility by comparing unprojected depths directly. Specifically, given depthmaps $D_1, D_2 \in \mathbb{R}^{H \times W}$, poses $T_1, T_2 \in \text{SE}(3)$, and camera projection functions $\pi_s, \pi_t$, we compute the corresponding projected pixel by:

$$p_s = T_1 \pi_1^{-1}(i_s) D_1(i_s), \quad i_t = \pi_2(T_2^{-1} p_s), \quad p_t = T_2 \pi_2^{-1}(i_t) D_2(i_t) \tag{S.2}$$

Since $i_s$ and $i_t$ are corresponding pixel locations, $p_s, p_t$ and target camera center $O_2$ are collinear. Note that we filter out-of-view or points behind the target camera when computing $i_t$ as non-covisible.

$$\|p_s - p_t\|_2 = |\|p_s - O_2\|_2 - \|p_t - O_2\|_2| = |\|p_s - O_2\|_2 - D_2(i_t)| \tag{S.3}$$

is the difference between the expected depth $\|p_s - O_2\|_2$ of the projected 3D point $p_s$ and the perceived depth $D_2(i_t)$ of the corresponding 2D pixel $i_t$ in the target camera.

Interpolating $D_2(i_t)$ from the discrete depthmap $D_2$ is vital to obtain a realistic covisibility mask. While $i_s$ is typically pixel-aligned — since we compute covisibility for source pixels in the target image — $i_t$ is derived from continuous depth and camera transformations, and thus almost always lies at a fractional pixel coordinate. We empirically found that bilinear interpolation yields better results than nearest-neighbor, as it provides a first-order approximation of the local depth geometry. In contrast, nearest-neighbor interpolation introduces heavy aliasing, especially on inclined surfaces. Although bilinear interpolation may produce ghosting artifacts, it is unlikely that a non-covisible pixel will match the expected depth closely enough to be mistakenly classified as covisible.

**Optical (Scene) Flow**  Unlike static scenes, optical flow datasets usually contain dynamic scenes and pairs in these datasets come from different timesteps. As the scene changes over time, determining covisibility requires scene-flow information to account for object motion. We build upon the formulation for static scenes and adjust the expected position with scene dynamics.

Formally, we use uniform camera projection model $\pi$ and all information as described in static scenes, optical flow ground truth $\phi^{gt} \in \mathbb{R}^{2 \times H \times W}$, and depth (disparity) change $D_{1 \to 2} \in \mathbb{R}^{H \times W}$. As optical flow describes how a source pixel $i_s$ moves to the target pixel $i_t$ in the image space, depth change details how the underlying 3D point changes in its depth. Specifically, the 3D point refered by $i_s$ at the source image with depth $D_1(i_s)$ would move to pixel $i_t = i_s + \phi^{gt}(i_s)$ in the target image, with an updated depth of $D_1(i_s) + D_{1 \to 2}(i_s)$. Thus, we can compute the source point in the target time and the projection error (similar to Eq. (S.3)) as:

$$p_{s,1 \to 2} = T_2 \pi^{-1}(i_t)(D_1(i_s) + D_{1 \to 2}(i_s)), \quad e = |\|p_{s,1 \to 2} - O_2\| - D_2(i_t)| \tag{S.4}$$

Then, we use the same interpolation and thresholding logic as the static datasets.

FlyingChairs is the only exception in this category, lacking both precomputed covisibility masks and scene flow information. Nonetheless, we include it during training to balance the relatively limited optical flow data compared to wide-baseline datasets. This does not pose a significant deviation from our covisibility-only training scheme due to the dataset's limited motion and relatively simple backgrounds. For correspondence training, we use the FoV mask as a proxy for the covisibility mask. We excluded FlyingChairs when supervising covisibility as explained in Sec. A.3.

**Rigid Posed Objects**  Rigid posed objects refer to scenes composed entirely of rigid objects whose poses are known at all timesteps. This setting can be seen as a special case of the scene-flow dataset where motion is fully defined between all pairs of timesteps. We adjust the expected position with the object movement information, similar to the formulation for optical flow.

Specifically, we assume all information in static scenes, the set of object poses $\{\tau_{1,2}^{(k)}\}_{k=1}^{K}$ at both time steps, $K$ being the number of objects, and $S : I \to \{1, \cdots, K\}$, the segmentation map that assign each pixel to the corresponding object ID. Given a source pixel $i_s$, we can obtain its object assignment $k = S[i_s]$ and its coordinate on this object as $\tau_1^{(k)-1}(T_1 \pi_1^{-1}(i_s)D_1(i_s))$. Since the object is rigid, the point will stay at the same object coordinate between source and target and be transposed to pose $\tau_2^{(k)}$ at the target frame. Combining them, we have:

$$p_{s,1 \to 2} = \tau_2^{(k)} \tau_1^{(k)-1}(T_1 \pi_1^{-1}(i_s)D_1(i_s)), \quad i_t = \pi_2(T_2^{-1} p_{s,1 \to 2}), \quad e = |\|p_{s,1 \to 2} - O_2\| - D_2(i_t)| \tag{S.5}$$

We threshold the error $e$ for covisibility as in the previous paragraphs.

**A.3. Covisibility Supervision Range.**  In addition to the covisibility mask, we compute a covisibility supervision mask that excludes regions where covisibility cannot be evaluated due to missing or invalid depth values. We apply supervision only within this mask to ensure accurate, though incomplete, training targets.

Formally, given depth validity masks $V_1$ and $V_2$ for the source and target images respectively, we first evaluate the validity of the target depth at the ground-truth flow locations as $V_{other}[i] = V_2(i + \phi_{gt}[i])$ and we obtain the covisibility supervision mask as

$$V_{covis} = (V_1 \cap \neg F_1) \cup (F_1 \cap V_{other})$$ (S.6)

where $F_1$ is the FoV mask, which is true for pixels in the source image whose corresponding 3D points have a valid projection into the image space of the second camera, regardless of occlusion. The first term captures the region that is out of view, while the second term captures the region that projects to the target's FoV and has valid depth at the target for confirming covisibility.

We used an all-zero covisibility supervision mask on the FlyingChairs dataset to avoid its approximated covisibility (actually FoV mask) from being used to train covisibility prediction.

## B Sampling Strategy

We explain our custom pair sampling strategy for the Scannet++V2 and Kubric4D datasets.

**ScanNet++ V2:** We compute all possible image pairs within each scene and retain those with sufficient covisibility. Specifically, following the procedure in Sec. A.2, we evaluate covisibility for all pairs of DSLR images in each scene and keep those with mutual covisibility greater than 25%.

**Kubric4D:** Kubric4D is the only dataset that enables sampling across both viewpoints and time. Accordingly, we bias our sampling toward pairs that involve changes in both dimensions. Specifically, Kubric4D has 2800 scenes with 16 fixed cameras in each scene and 60 frames per scene. We sampled 3600 pairs per scene with viewpoints and time change independently:

We aim for $60° - 90°$ angle difference for viewpoints. To achieve this, we first computed the rotation angle between all pairs of camera and assigned weight as

$$w(\alpha) = \begin{cases} 1 + \alpha, & \alpha \in [0, \pi/3) \\ 1 + \pi/3, & \alpha \in [\pi/3, \pi/2) \\ 0, & \alpha \geq \pi/2 \end{cases}$$ (S.7)

We sample frame differences to bias toward large difference since motion in Kubric4D is small. Specifically, we sample frame difference between 0 and 40, with probabilities increasing linearly such that the largest frame difference has twice the probability of being selected compared to the smallest. Given a sampled difference, we then uniformly choose a valid start and end frame.

## C TA-WB Training & Testing Dataset

TartanAir provides images covering all six directions around each scene, enabling us to design a geometric sampler that explicitly controls viewpoint differences when sampling covisible pairs.

**Geometric Sampler** The geometric sampler generates pairs of rendering directions and source–target cameras based on geometric constraints for viewpoint difference and coarse covisibility check. An overview is presented in Fig. S.1.

We first voxelize the scene and compute the set of visible voxels for each camera. The sampling process begins by randomly selecting a source camera center and a visible voxel nearby, establishing the viewing direction for the source. Based on this direction, we identify candidate target cameras whose viewing angles differ by the desired amount. Then, we filter out candidates that cannot see the selected voxel based on pre-computed covisibility. In this way, we are able to sample covisible yet geometrically controlled viewing directions. Finally, we sample a random roll angle from $\mathcal{N}(0, 0.1)$ to complement the viewing direction into a rotation, and apply a random perturbation to all axes from $\mathcal{N}(0, 0.1\mathbf{I}_3)$. These perturbations prevent the sampled viewing direction from always focusing on the voxel center, adding diversity to the sampling.

After rendering the images, we do additional filtering to ensure their quality. We filter out pairs with any of their images containing more than 10% of over- or under-exposed pixels, and if any of the

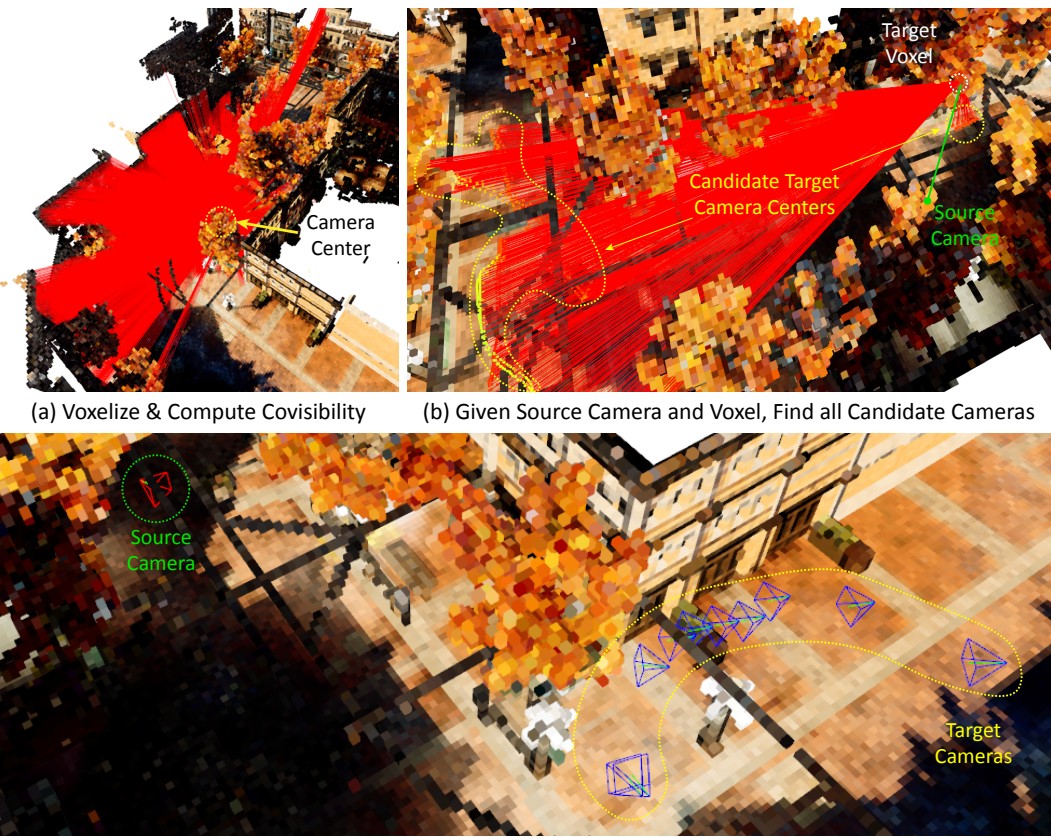

(a) Voxelize & Compute Covisibility  (b) Given Source Camera and Voxel, Find all Candidate Cameras

(c) Filter by Covisibility & Sample

Figure S.1: **The Geometric Sampler**: (a) From the pointcloud of a scene, we voxelize it and compute the covisibility between all camera centers and all voxels. (b) We randomly select a camera location as the source camera and a target voxel for the source camera to center at. We filter out all candidate camera position that forms a required viewpoint difference when looking at the same target voxel. (c) We filter out candidate cameras by covisibility.

forward/backward covisibility is less than 20%. We further check if the pair is solvable, i.e., does the pair provide enough visual evidence to establish a match? To do this, we warp the target image according to the ground-truth label (similar to Fig. 1, 4) and try to match it to the source image with Superpoint + Lightglue [10, 38, 49]. Since warping is done with ground truth, the matcher should ideally return near-zero pixel displacement in covisible regions. If it does not, the pair lacks enough information to support matching. We retain only pairs with an average matching error below 6 pixels.

**TA-WB Benchmark**  We use the geometric sampler to select pairs from the `OldScandinavia`, `Sewerage`, `Supermarket`, `DesertGasStation`, and `PolarSciFi` environments in TartanAirV2 [68]. We sample approximately equal numbers of pairs from the angular bins $[0°, 30°]$, $[30°, 60°]$, and $[60°, 90°]$, and allocate roughly half as many pairs to the $[90°, 120°]$ bin. Samples of the dataset are provided in Fig. S.2.

## D   Training the Refinement

We trained the refinement module separately, using a frozen base model obtained from the initial training stage. Since the refinement value is computed via attention between the source pixel feature and features in a local neighborhood around the regressed flow target, it can be interpreted as a multi-modal distribution centered around the base model's predicted flow. We use the cross-entropy loss to supervise the distribution at the ground-truth location. Importantly, we limit supervision to pixels whose ground-truth flow falls within the $7 \times 7$ neighborhood and use a softened target. Rather than having the nearest pixel that is closest to the flow target as a classification target, we distribute

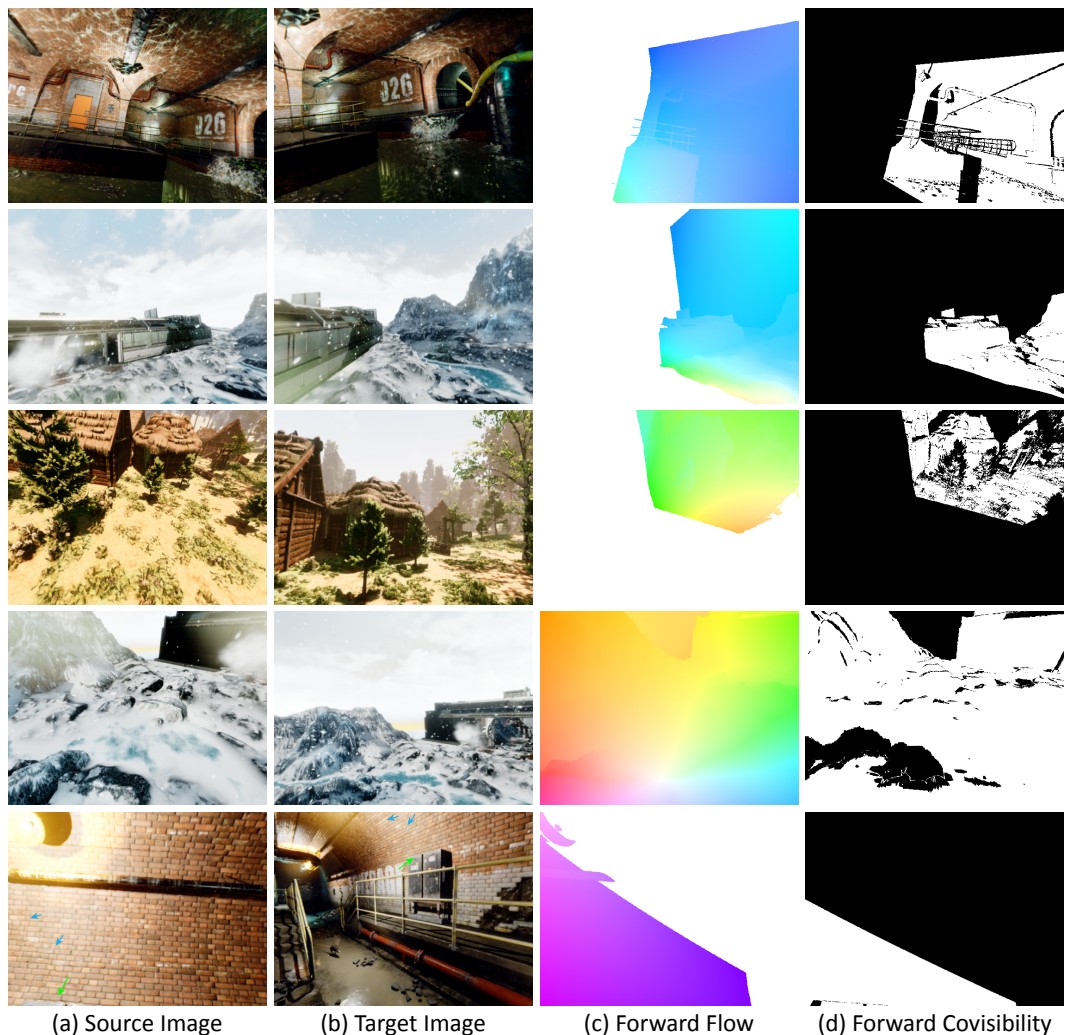

|  |  |  |  |
|---|---|---|---|
| (a) Source Image | (b) Target Image | (c) Forward Flow | (d) Forward Covisibility |

Figure S.2: **Example Images from TA-WB Benchmark**: The benchmark contains dense correspondence annotation and accurate covisibility for challenging viewpoint shifts.

$$w_1 = (1-\alpha)\beta$$
$$w_2 = \alpha\beta$$
$$w_3 = (1-\alpha)(1-\beta)$$
$$w_4 = \alpha(1-\beta)$$

Figure S.3: **Refinement Target Weights**: Given an inlier ground-truth flow target, we obtain its adjacent pixels and assign a continuous weight based on the sub-pixel location $(\alpha, \beta)$ of the target.

smooth weights across the four adjacent pixels, with values that change continuously based on the flow target. We found that such a target is easier to train and enables sub-pixel refinement. The weights are shown in Fig. S.3.

We trained the refinement module on the BlendedMVS, MegaDepth, Habitat, and ScanNet++V2 datasets using image pairs as listed in Table 1. We selected these datasets due to their relatively high sub-pixel accuracy. The base model was frozen during this stage, and the refinement module was

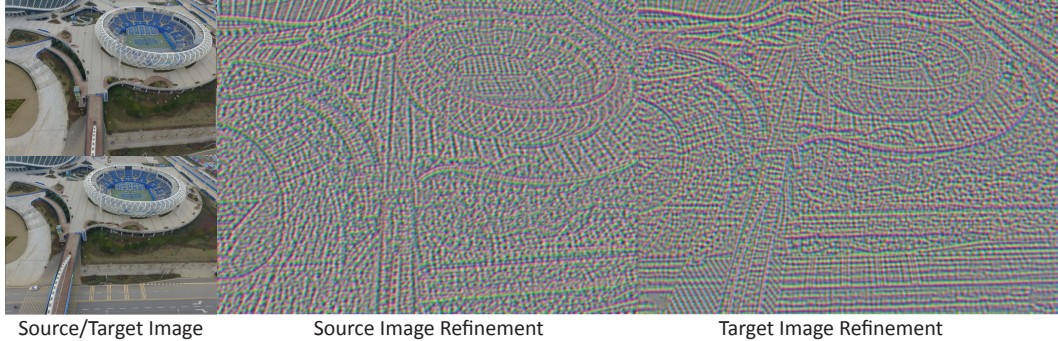

| Source/Target Image | Source Image Refinement Features (PCA) | Target Image Refinement Features (PCA) |

Figure S.4: **Example of Refinement Features**: We visualized the refinement features for a pair of images with PCA. The features exhibit emergent high-frequency and edge-following behavior.

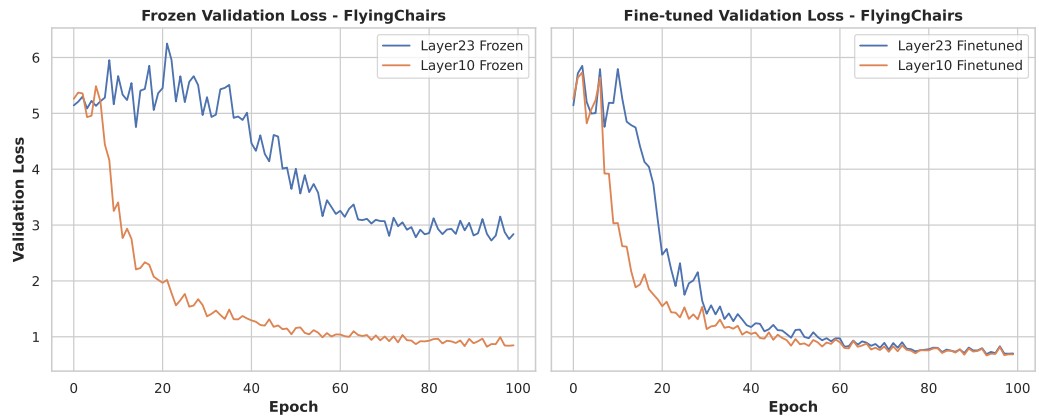

Figure S.5: **Freezing DINOv2 encoder is suboptimal when training UFM on FlyingChairs**: We show the validation EPE of FlyingChairs using features from different layers of a frozen pre-trained encoder (left) and finetuning the pre-trained encoder truncated to a specific layer (right).

trained for 30 epochs with a learning rate of $1 \cdot 10^{-4}$. All other optimizer settings are the same as the 560 base model training, as detailed in Section 3.4. Fig. S.4 shows a visualization of the trained features, where we see high-frequency and edge-following behavior that encodes the local details.

## E Effect of Freezing the Encoder

We found that freezing the DINOv2 encoder and using its last-layer features was suboptimal for UFM. Specifically, when training UFM on the FlyingChairs dataset, we observed a significant validation EPE gap between using features from the last layer versus intermediate layers of the frozen DINOv2 encoder. As shown in Fig. S.5, UFM trained with the last layer features from frozen DINOv2 obtained near 3 EPE, whereas features from layer 10 yielded sub-pixel performance. This gap is not observed in the finetuned setting, given sufficient training.

**E.1. Hypothesis for Performance Gap with Frozen Features:.** The task of predicting the dense correspondence can be roughly divided into 3 steps. For a patch in the source image, it would need to: (1) understand the content in its own patch, (2) find the corresponding patch(es) in the other image, and (3) copy its coordinate difference. While one may argue that step (2) is unnecessary because the network can leverage structural priors or surrounding context to fill in the gap, it remains the most direct and reliable route to accurate correspondence due to the causal nature of the task.

Step (2), i.e., finding the corresponding patch(es) in the other image, is achieved in only one structure of UFM - the global attention. This is because all other components either project patch features independently or operate solely on tokens from a single image, lacking direct cross-image interaction.

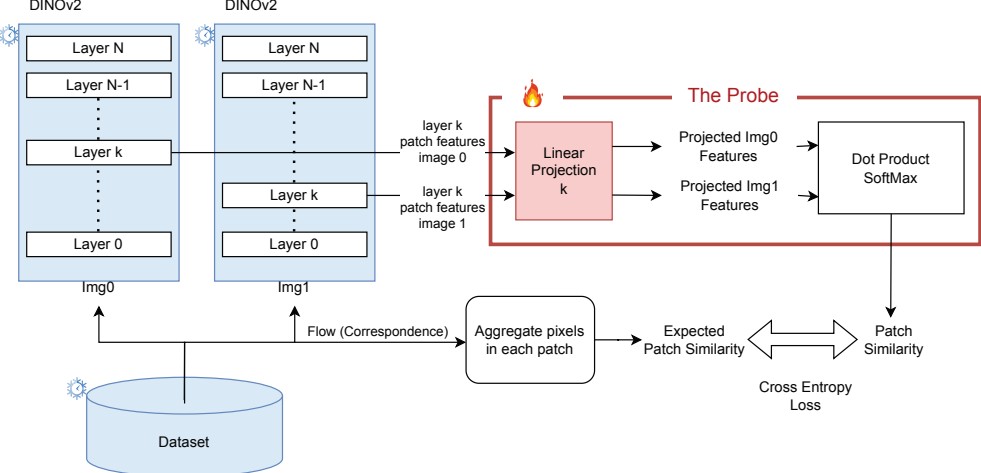

Figure S.6: **Setup for the Probing Experiment**: For each layer in a frozen image encoder, we extract patch features for a pair of images and apply a shared linear projection. Softmax attention is computed between source and target features, and the resulting similarity distribution is compared to ground-truth correspondences via cross-entropy loss. The final training loss serves as a proxy for correspondence information encoded at each layer.

In the global attention module, (2) is realized by the attention computing, which depends on the dot-product similarity of the patch feature after a learnable linear projection.

This implies a key requirement: *Patch features must encode information that reveals their correspondence, or "corresponding features", such that they attend selectively to their corresponding patches in the other image after a simple linear projection.* We designed a probing experiment to quantify the upper bound of the corresponding features in each layer of a frozen encoder, and later establish its correlation to UFM's performance experimentally.

**E.2. Probing Experiment:.** We overfit a simple network on top of a trained backbone to a specific dataset, using the converged training loss as a proxy for the presence of relevant information in the backbone representations. It was used as an analysis strategy in NLP as training "probing classifiers" to associate the internal representation of the model with explicit properties [3]. We use a similar probing experiment to test the presence of corresponding features from layers in a frozen DINOv2.

The outline of our probing experiment is shown in Fig. S.6. We select a relatively small dataset and disable all augmentation to ensure that training will converge. We infer each pair of images through the frozen DINOv2 encoder and project the source and target features through a layer-specific linear layer. We then compute softmax dot-product similarity to mimic the global attention mechanism. Each layer's probe is trained independently, and its performance reflects how well the layer encodes corresponding features that can be revealed during the global attention. Patch-wise similarity is defined as the proportion of pixel-wise correspondences between patches, weighted by covisibility. Formally, given correspondence and covisibility labels $\phi^{gt}, C^{gt}$, the ground-truth patch similarity $s(P_s, P_t)$ between a source patch $P_s \subset I_1$ and target patch $P_t \subset I_2$ is defined as:

$$s(P_s, P_t) = \frac{\sum_{i \in P_s} 1(i + \phi^{gt}[i] \in P_t) \cdot C^{gt}[i]}{|P_s|} \tag{S.8}$$

Given a fixed dataset, we infer a pair of images through the frozen image encoder and obtain the patch features at all layers. For each layer, we project the source and target features using a shared linear layer and compute their softmax attention, resulting in a binary distribution of pair-wise patch similarity. This predicted distribution is then compared to the ground-truth similarity using a cross-entropy loss. We train only the projection layers on this dataset and use the final training loss as an indicator of how well the features at each layer encode correspondence information.

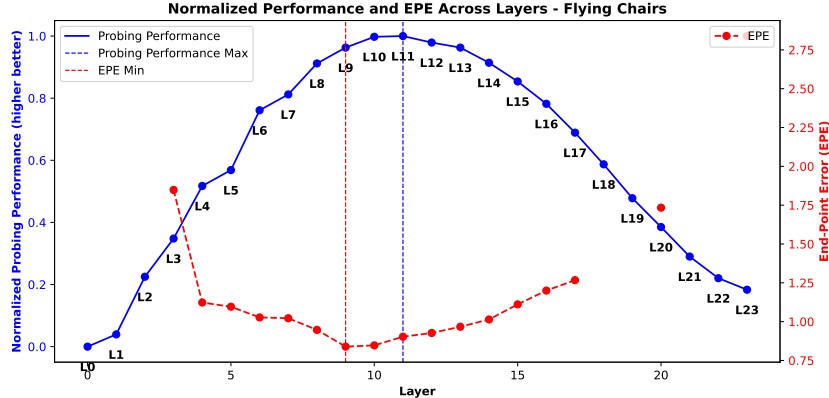

Figure S.7: **Correlation between probing and val. EPE**: We plotted the probing performance (blue) and the EPE of UFM on FlyingChairs when using frozen DINOv2 features from different layers.

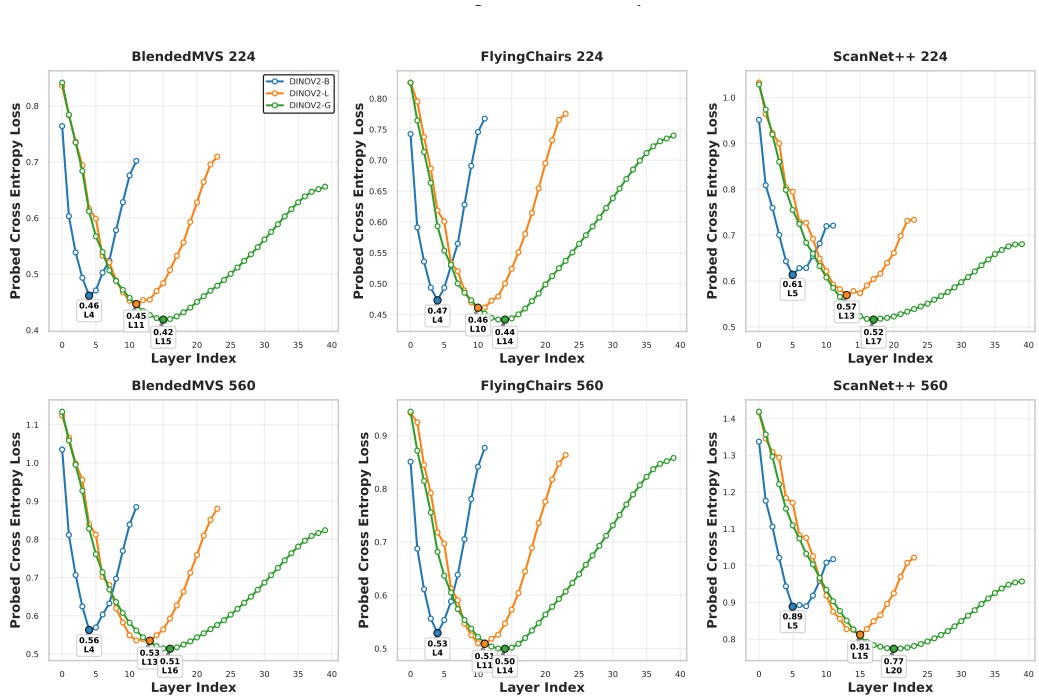

Figure S.8: **Consistent probing results** on other datasets, resolutions, and encoder sizes showing that the last layer from DINOv2 does not provide the best corresponding features and performance.

**E.3. Probing Results using Correlation:.** To test whether the cross-entropy loss from probing correlates with EPE performance, we trained UFM using different frozen layers of DINOv2 on the FlyingChairs dataset and collected their loss in the probing experiment. We normalized the cross-entropy loss value into probing performance between $[0, 1]$. According to Fig. S.7, we found a strong correlation between the loss in the probing experiment and the final validation EPE, and the peaks differ only by 2 of the total 24 layers. This suggests that, for UFM, probing performance may serve as a reliable indicator for selecting effective feature layers. Furthermore, this supports the hypothesis that the last layer of DINO does not provide the strongest corresponding feature, thus leading to suboptimal performance. We further show additional probing results on other datasets, resolutions, and DINOv2 encoder sizes in Fig. S.8. We found a consistent trend where the intermediate layers encode stronger correlating features and perform better.

Table S.2: **Matching Head Ablation**: With identical backbone and training data, a regression-based flow head (UFM design) using DPT achieved higher precision than a patch-wise matching head from UniMatch.

| Head Type | Sintel-C | Sintel-F | KITTI | DTU | ETH3D | TA-WB |
|---|---|---|---|---|---|---|
| | | | EPE [covis] | | | |
| Patch Matching + Upsample | 0.91 | 1.49 | 2.05 | 5.36 | **5.47** | 13.85 |
| Regression (ours) | **0.79** | **1.44** | **1.87** | **2.64** | 5.56 | **12.87** |

## F  Discussion on the use of Matching in UFM Architecture

As many prior works have shown the benefits of using matching for correspondence prediction, we would like to further discuss and ablate our choice to regress the initial flow combined with applying matching only at pixel resolution, not patch resolution.

**Matching at the patch level**    Matching in patch-level aggregates subsets of pixels as a patch, then matches them with other patches to produce an initial flow hypothesis at low resolution. The initial coarse flow will be upsampled and refined later, following the "coarse-to-fine" paradigm. To better present our findings, we would first like to clarify two possible meanings of the term "coarse-to-fine": (1) the predicted flow is upsampled from a coarse to a fine spatial resolution, or (2) a rough prediction is progressively refined for higher accuracy. While prior methods often combine both, our approach uses (2) but not (1).

To support our choice, we replaced the DPT head of UFM $_{560}$ with the UniMatch matching head and trained with the same protocol as described in Section 3.4. We adopt this head as a representative of patch-scale matching methods. Hence, the matching head now benefits from the same backbone capacity and unified dataset as UFM $_{560}$. The only differences are the final flow prediction head and how the flow loss is applied. To maintain equivalence in supervision, we modified the UFM robust loss to match how UniMatch applies its loss - supervising on both the final flow and the bilinearly upsampled coarse flow. Essentially, the UniMatch head first performs patch-level (coarse) matching, which is then convex upsampled to pixel-wise correspondence. We follow their original codebase and refer to formulas (1) - (7) from GMFlow [73] for the exact definition of the UniMatch head.

We evaluated them on covisible end-point-errors and report the results in Table S.2, where we observed that the UniMatch Head yields a 22% increase in EPE on average. Hence, we conclude that the regression head is more accurate in predicting correspondence and scales better to the combined dataset, corroborated by evidence in Fig. 5.

**Matching at the pixel level**    Direct global matching of pixel descriptors allows MASt3R to achieve high precision even for large motions. Using a similar architecture and dataset scale, we show that the regression paradigm in UFM $_{560}$ achieves competitive EPE performance on challenging wide-baseline datasets, where the average flow reaches about 90 - 200 pixels. We attribute this to UFM's superior *supervision efficiency* compared to MASt3R. While MASt3R trains with at most 4096 correspondence pairs per frame, UFM leverages supervision from all covisible pixels. This dense supervision is a key factor that allows UFM to achieve similar or better accuracy with much higher speed than MASt3R.

**Matching and Regression are not mutually exclusive**    Through UFM-Refine, we demonstrate that matching-based local refinement can be effectively combined with a regressed initial hypothesis to further enhance accuracy. This forms a key distinction between UFM-Refine and previous dense correspondence approaches. Only with an efficient and accurate regressed initialization can one confidently narrow the search range for pixel-wise matching, thereby achieving both the efficiency of regression and the precision of pixel-wise matching.

## G  Additional Evaluations

We provide additional zero-shot evaluations of UFM on Sintel and KITTI test sets in Table S.3, S.4. While we still observe UFM outperforms UniMatch under zero-shot settings (neither method was trained on Sintel or KITTI), we observed UFM is biased towards static objects. As in Table S.4, UFM is significantly better in the background than in the foreground objects, which are mostly

Table S.3: **Zero-Shot Performance on Sintel Test Set:** UFM outperforms UniMatch in the overall metric and most sub-items, except for sub-items on Sintrel-Clean.

| Dataset | Method | EPE all | EPE matched | EPE unmatched | d0-10 | d10-60 | d60-140 | s0-10 | s10-40 | s40+ |
|---------|--------|---------|-------------|---------------|-------|--------|---------|-------|--------|------|
| *Sintel-Final* | UFM 980 - refine | **3.28** | **1.75** | **15.80** | **3.28** | **1.27** | **1.21** | **0.69** | **2.07** | **19.07** |
| | UniMatch-CT | 4.14 | 1.97 | 21.86 | 3.77 | 1.59 | 1.28 | 0.89 | 2.43 | 24.40 |
| *Sintel-Clean* | UFM 980 - refine | **1.68** | 0.72 | **9.52** | 1.54 | 0.67 | 0.43 | 0.48 | 1.23 | **8.71** |
| | UniMatch-CT | 1.80 | **0.57** | 11.85 | **1.18** | **0.45** | **0.40** | **0.37** | **0.93** | 11.03 |

Table S.4: **Zero-Shot Performance on KITTI Test Set:** UFM outperforms UniMatch in the overall metric and background objects, but not foregrond objects, which are mostly dynamic.

| Method | Range | F1-bg | F1-fg | F1-all |
|--------|-------|-------|-------|--------|
| UFM 980 - refine | All | **8.67** | 22.08 | **10.9** |
| UniMatch-CT | All | 18.01 | **17.27** | 17.89 |
| UFM 980 - refine | Not occluded | **4.93** | 19.6 | **7.59** |
| UniMatch-CT | Not occluded | 9.75 | **14.61** | 10.63 |

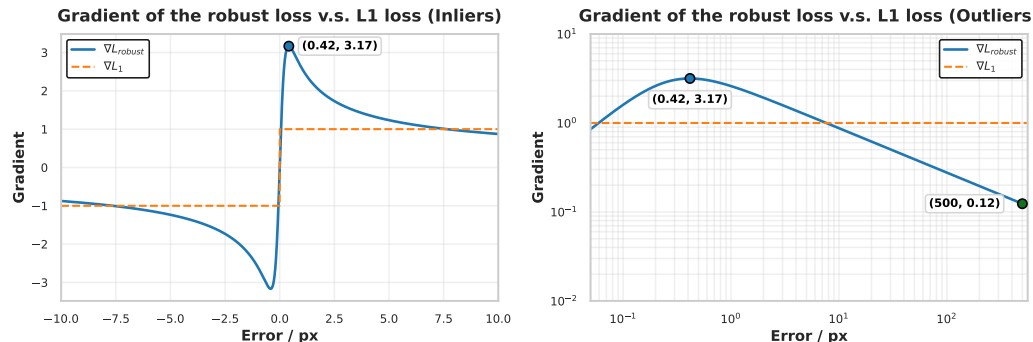

Figure S.9: **Gradient of the Robust Charbonnier Loss:** Compared to $L_1$ loss, the robust loss used by UFM puts higher gradient to the inliers (peaks at error of 0.42 pixel), while maintaining non-negligible (0.12 at 500 pixel error) gradient for even the largest outliers to slowly correct them.

dynamic. Future work may include more sceneflow datasets, such as ParallelDomain [63], to balance the distribution.

# H   Visualization on the Robust Charbonnier Loss

We provide an additional visualization of the robust Charbonnier loss. The loss is defined as:

$$L_{robust}\left(x; \alpha, c\right) = \frac{|\alpha - 2|}{\alpha}\left(\left(\frac{\left(\frac{x}{c}\right)^2}{|\alpha - 2|} + 1\right)^{\frac{\alpha}{2}} - 1\right) \tag{S.9}$$

We followed RoMa's choice of $\alpha = 0.5$, $c = 0.24$. As shown in Fig. S.9, this combination of $\alpha$ and $c$ puts a high gradient on inliers while maintaining non-negligible gradient for even the largest outliers to slowly correct them.

