# OpenReview forum: "UFM: A Simple Path towards Unified Dense Correspondence with Flow"
_NeurIPS.cc/2025/Conference — NeurIPS 2025 poster_

### Official Review · Reviewer_NLte · 2025-06-19

**Clarity:** 4
**Significance:** 2
**Originality:** 2
**Rating:** 5
**Confidence:** 4

**Summary:**

The authors propose a model and set of training datasets for the task of dense correspondences, unifying optical flow and dense feature matching.
The coarse part of the model is a two-view transformer built on top of DINOv2-L features, which is trained to output a corrrespondence field and a covisibility mask for each token position. If needed, these correspondences can be refined by a lightweight refiner on top.
On the data-side, the authors compile a set of datasets similar to that of MASt3R, extracting correspondence either from depthmaps or from optical flow.
The experiments show that the method, UFM, performs competivetely with specialized models, while being significantly simpler and faster.

**Questions:**

1. I do think I entirely understood the proposed loss. Is the only supervision for the flow the robust Charbonnier loss? In RoMa this was used on top of a classification type loss. Would only using this loss not cause the network to get very low gradients when matching fails? I would be curious to hear the authors explain the thought process a bit more.

2. W.r.t. my comments regarding comparisons to MASt3R, I would be interested in hearing an argument from the authors why their full method is needed, when MASt3R already seems to perform well (and possibly outperforms the proposed method). For example, why not simply train a small regression head on top of the MASt3R decoder?

3. W.r.t to MD1500, I am interested in hearing why the authors have not included this very standard benchmark.

Currently my rating is around a borderline, with my main concern being MASt3R comparisons. Given a convincing rebuttal to the questions above I would be willing to raise my rating to accept.

**Ethical Concerns:**

["NO or VERY MINOR ethics concerns only"]

**Final Justification:**

The rebuttal clarified the loss, and the added ablation is a nice validation of the approach. W.r.t. comparisons to MASt3R, I think that while the comparisons w.r.t. time are not entirely fair, building dense matching efficiently on top of MASt3R would require a substantial rewrite and is perhaps not a fair comparison from my part. The performance of the proposed method is additionally significantly higher than MASt3R on the WxBS benchmark, indicating that there is a clear usecase, beyond just more dense predicitions.

I also read the discussions with the other reviewers, and believe that their concerns were adequately addressed.

I have thus increased my rating from 4->5.

**Limitations:**

yes

**Quality:**

3

**Strengths And Weaknesses:**

**S1 - Simple model, strong results:** The method outperforms matching methods such as RoMa and optical flow methods such as FlowFormer. While technically not SotA on optical flow tasks, as later methods outperform FlowFormer, I believe the performance is satisfactory. I was also positively surprised to see that the authors made the effort to compare their approach to these previous methods when trained on the same data (see Figure 5), which validates their modeling choice (and also their 2nd contribution).

**S2 - Unifying flow and wide-baseline matching:** For downstream tasks where these types of models are used (both in academia and industry), it is incredibly beneficial to have a model that is not overly sensitive to the specifics of the setup. While RoMa is robust to large temporal changes, it is not particularly accurate for thin and small objects. The proposed model takes a step towards a setting where one can simply input any image pair and expect to get correct correspondences in a timely manner, which I think is an important contribution.

**W1 - Unclear robustness to temporal/etc variations in images:** As the authors mention on l.285-l.292, finetuning the DINOv2 backbone tends to reduce generalization to more semantic types of matching. I interpret this to mean that performance on more difficult feature matching benchmarks such as MegaDepth-1500, and WxBS, decrease due to this choice. Unfortunately the authors do not provide results on these benchmarks, particularly MD1500 would be relevant as it is the standard feature matching benchmark.

**W2 - Unclear improvement in comparison to MASt3R:** Looking at Table 2 and Table 3, MASt3R generally outperforms the proposed method (although MASt3R can be improved by using the proposed refiner). While MASt3R is significantly slower, I would assume that most of the runtime comes from matching its features in a brute-force manner. As this matcher is very basic, I'm not entirely convinced that the claimed 60x speedup is representative.

---

> ### Author Rebuttal · Authors · 2025-07-31
>
> Thank you for your thoughtful and supportive comments. We are encouraged that you find UFM to be **simple, efficient, and delivering strong results**. We value your recognition of **our effort to compare prior methods on the same data**, which, as you noted, **substantiates our architectural choice**. We value your endorsement that **UFM presents an important step towards application** for its robustness to setup variations.
>
> We added WxBS results to show semantic matching capability. We believe that comparison to MASt3R might not have been very clear in the paper and would like to clarify. We further explained our motivation for the robust loss.
>
> - **Q1: Unclear robustness to temporal/semantic variations in images. Why not include the MegaDepth 1500 benchmark?**
>
>     **Because we train exhaustively on the MegaDepth dataset. We add the WxBS Experiment and show UFM is reasonably robust for extreme conditions.**
>
>     We evaluated our model under extreme conditions using the WxBS dataset, which includes low-light scenes, multiple visual spectrums (e.g., IR to RGB), day-night transitions, seasonal changes, and scale changes. We report performance on fundamental matrix estimation using PCK, i.e., the percentage of pre-labeled keypoints consistent with the prediction under an epipolar error threshold (higher is better, 10px is commonly used).
>
>     | Thresholds | 3px  | 5px  | 10px | 15px | 19px |
>     |------------|------|------|------|------|------|
>     | RootSIFT   | 0.06 | 0.08 | 0.10 | 0.13 | 0.16 |
>     | Superpoint + SuperGlue | 0.26 | 0.33 | 0.46 | 0.51 | 0.55 |
>     | LoFTR      | 0.48 | 0.57 | 0.67 | 0.71 | 0.73 |
>     | RoMa       | **0.74** | **0.85** | **0.92** | **0.96** | **0.96**|
>     | MASt3R     | 0.32 | 0.42 | 0.56 | 0.62 | 0.65 |
>     | UFM_560    | 0.49 | 0.63 | 0.74 | 0.77 | 0.78 |
>     | UFM_560-Refine | 0.49 | 0.62|  0.75 | 0.79 | 0.81 |
>
>     As already noted in Section 5 (Limitations), compared to RoMA (the best performing one, where DINOv2 is frozen), finetuning the DINOv2 encoder in UFM improves precision but reduces semantic matching capabilities (as also supported by our supplementary material analysis, Section E). Despite this, UFM ranks second on the WxBS benchmark with a 0.74 PCK@10, showing strong performance and outperforming MASt3R under extreme conditions.
>
> - **Q2: Comparison to MASt3R is unclear**
>     - **Q2.1: Looking at Table 2 and Table 3, MASt3R generally outperforms the proposed method (in precision), although MASt3R can be improved by using the proposed refiner.**
>
>         **We believe that you are only comparing our base model to MASt3R**, since the refinement model holds **8/12** advantageous metrics in Table 2, and at least **5/9** metrics in Table 3.
>
>         **We wish to highlight an important detail that biases the comparison in Table 2 in favor of MASt3R - UFM is actually achieving comparable precision on MASt3R's most confident pixels, a more demanding evaluation setup for UFM.**
>
>         As explained in lines 236-239, MASt3R is a sparse method because its feature matcher can only output correspondence that **passed its cycle-consistency check**. **By evaluating UFM on MASt3R’s set of output pixels, we give MASt3R an advantage in selecting its most confident matches**. Statistically, **MASt3R skips 2.1%, 2.0%, and 10% of the samples almost entirely** (defined as reporting < 10% of covisible pixels) in DTU, ETH3D, and TA-WB. Thus, while the numbers of the base model in Table 2 may only appear slightly inferior, UFM is achieving comparable precision.
>
>         We agree that incorporating the proposed refiner, specifically the U-Net and sub-pixel weighted average refinement, would enhance MASt3R’s performance. However, we would like to emphasize that **these components are integral to UFM’s design, specifically introduced to improve the precision for inliers, as motivated in lines 153–154.** This function is further supported by the significant reduction in 1px outlier rate shown in Table 2 when comparing UFM with and without refinement.
>
>     - **Q2.2: The claimed 60x speedup (in table 2) is not representative, as most of MASt3R’s runtime is on the matching process, and its matcher is very basic.**
>
>         **While we agree that the 60x speedup is not fully accurate, UFM still offers a significant speed advantage in dense applications.** Indeed, we should use the runtime of our refinement model for comparable performance in Table 3. While we acknowledge that a significant portion of MASt3R’s runtime comes from the feature matching stage, we do not characterize the matcher as “very basic,” since one of MASt3R’s core contributions is its fast reciprocal matcher. That said, we also found the follow-up paper *Speedy MASt3R*, which reports a 54% runtime reduction through various optimizations. **Taking these updates into account, we would revise the reported speedup to approximately 24×, which is still significant.**
>
>     - **Q2.3: Why UFM is needed, when MASt3R already seems to perform well.**
>
>         **UFM holds the following advantages in applications:**
>         1. **Completeness**: UFM outputs dense correspondences, while MASt3R’s output is usually incomplete. Hence, UFM can be applied to tasks like image warping, while MASt3R cannot.
>         2. **Efficiency**: When dense outputs are required, UFM has a significant speed advantage (as revised in Q2.2) that makes it favorable to real-time applications like SLAM.
>         3. **Semantically more robust**, as shown in Q1.
>         4. **Sub-pixel Matching**: MASt3R's proposed matcher predicts integer coordinates only, lacking sub-pixel prediction. UFM and its refinement provides sub-pixel predictions.
>         5. UFM also provides an **explicit covisibility estimate**, which is useful for tasks such as loop-closure.
>
>     - **Q2.4: Why not simply train a small regression head on top of the MASt3R decoder?**
>
>         Firstly, we aim to modify the backbone to **incorporate advantageous components for correspondence**.
>
>         1. We replaced CroCo with DINOv2 **for its semantic matching ability**, justified by Q1. Finetuning MASt3R may not bring robustness for semantic matching as our dataset contains mostly geometric variations.
>         2. We have **simplified MASt3R’s transformer by removing RoPE** (which we found not required).
>         3. We **accelerated it** by replacing cross-attention with global attention, enabling **better performance with Flash Attention** at long sequence lengths without increasing theoretical compute (see lines 142–145).
>
>         More fundamentally, we want to **simplify and reduce the time to train a correspondence foundation model from scratch**. UFM training takes 4 days at 8 H100, while MASt3R’s training depends on 6.5 days for DUSt3R and 11 days for MASt3R under 8 A100 GPUs, which is still much longer after considering the hardware differences.
>
>
> - **Q3: Thought process on the robust loss.**
>
>    Our goal is to **increase the precision of the inliers instead of having mediocre predictions for all pixels**. For tasks like pose estimation, which rely on a highly accurate subset of all the correspondences, we found it correlates best with fine inlier rates instead of EPEs.
>
>
>     **We have confirmed this improvement by finetuning a previously trained checkpoint** (not reported in paper) with robust vs. L1 loss under the same setting.
>
>     ||DTU EPE | 1/2/5 px ouliers | ETH3D EPE | 1/2/5 px outliers | ScanNet++ EPE | 1/2/5 px outliers|
>     |-|-|-|-|-|-|-|
>     |L1|6.76|72/46/17|**3.63**|55/30/11|5.31|70/43/17|
>     |Robust|**6.17**|**64/38/15**|3.99|**46/26/10**|**4.93**|**66/39/16**|
>
>     As shown, the robust loss improves both inlier rates and even most EPE compared to the L1 loss. We attribute this to two factors: (1) the dataset includes unsolvable pairs that are better ignored by the robust loss, and (2) the robust loss promotes easy-to-hard learning.
>
>     - **Q3.1: Is the only supervision for the flow the robust Charbonnier loss?**
>
>         Yes.
>
>     - **Q3.2: Would only using this loss not cause the network to get very low gradients when matching fails?**
>
>         **While the gradients are comparatively smaller for large outliers than for inliers, they are not entirely negligible, even for the largest outliers.** The robust Charbonnier loss (with c=0.24,a=0.5) has gradients that peak at 3.16 near inliers (e.g., x=0.4) and remain non-zero (e.g., 0.12 at x=500) for large outliers. This ensures strong updates for inliers while still allowing meaningful correction for learnable outliers. In early experiments, we tried training from scratch with a L1 vs robust loss (under a slightly different setting) and observed no significant difference in convergence speed.

---

> > ### Comment · Reviewer_NLte · 2025-08-01
> >
> > Dear authors,
> >
> > Wrt. Q1, while unfortunate that the MegaDepth-1500 benchmark was included in the training set, I appreciate that for that reason it was not included as a benchmark.
> >
> > Wrt. comparisons to MASt3R, I somewhat agree, particularly with the comparison on WxBS. Of course, on that benchmark RoMa does outperform the proposed method, but RoMa is loses most other comparisons. I think the inclusion of WxBS in general is good, as it shows the advantage of the different methods, which might help guide future research.
> >
> > Regarding the robust loss, the provided experiment is interesting, and perhaps also mirrors learnings from optical flow (I seem to recall methods like GMFlow also only using robust loss). I do think the combination of classification/correlation-type losses as in SuperGlue/LoFTR/RoMa with robust regression losses for refinement, is still probably better than purely robust regression, but perhaps I am wrong. It certainly makes the loss simpler, which is a plus.
> >
> > I will likely increase my score to accept.

---

> > > ### Author Response · Authors · 2025-08-03
> > >
> > > Thank you for your additional comments and suggestions. We’re glad the inclusion of WxBS and the discussion on loss functions were helpful.
> > >
> > > We will include the WxBS result in the revision to evaluate UFM better and support our limitations section. We appreciate your insight on combining correlation-type loss with robust regression. This was one of our early explorations, using patch correlation as auxiliary supervision. We tried it on a small scale and applied it in different places of the network, but did not see improvements. We suspect factors such as data scale, loss weighting, and where the loss is applied in the network may affect its impact. Nonetheless, we see this as a promising direction and will investigate further in future work.

---

### Official Review · Reviewer_RKdL · 2025-06-23

**Clarity:** 3
**Significance:** 3
**Originality:** 3
**Rating:** 5
**Confidence:** 5

**Summary:**

This paper proposes a unified model for wide-baseline feature matching and optical flow, which are usually studied separately in previous works. Thanks to the unified formulation, the proposed model can be trained on combined datasets from both tasks, where the performance is better than separately trained models. Technically, the proposed model is mainly a large Transformer-based architecture, with a flow head to regress pixel displacements and a covisibility head to predict covisibilities across two input views. Experiments are conducted on wide-baseline feature matching and optical flow datasets, the proposed method achieves comparable or better results than previous specialized methods, but with a unified model.

**Questions:**

The key questions are listed in the Weaknesses.

In addition, I think another related work that is worth mentioning and discussing is: [Perceiver IO: A General Architecture for Structured Inputs & Outputs. ICLR 2022], which also tried to develop a unified model for different tasks and also reported the experiments on optical flow.

In general, I like the idea of this paper. However, I am currently not fully convinced by the experiments (detailed in Weaknesses). I am happy to raise my rating if these questions are well-addressed.

**Ethical Concerns:**

["NO or VERY MINOR ethics concerns only"]

**Final Justification:**

My initial concerns are well-addressed, and thus I vote for accept.

**Limitations:**

yes

**Paper Formatting Concerns:**

No major formatting issues.

There is one typo in Line 9: "find" -> "fine".

**Quality:**

3

**Strengths And Weaknesses:**

### Strengths

- The unified model is appealing since the two tasks of wide-baseline feature matching and optical flow indeed share a lot of similarities.
- One benefit of the unified model is that it can be trained on the combined datasets. In Table 5, this paper shows that the combined-dataset training leads to mutual improvement on both tasks.
- To train the unified models, this paper compiles 12 datasets, with detailed preprocessing steps thoroughly discussed. These efforts provide a valuable resource to the community.

### Weaknesses

- This paper argues about the difference with previous coarse-to-fine methods in several places (e.g., Line 9 and Line 30), but eventually the proposed model still has a coarse-to-fine structure (with an additional refinement stage). I think this paper would like to highlight the difference in the "coarse" stage (global vs. local), instead of the overall coarse-to-fine structure in previous methods. In addition, I think the argument on global attentions used in the model is not well-supported by experiments. In Figure 4, some comparisons with RoMa are shown, but there are many differences in implementation and training datasets between the proposed method and RoMa, which makes it not easy to understand where the performance gains come from. A more rigorous comparison could be replacing the global attentions used in the proposed model with local attentions and train with the same setup and then compare their performance. Such an experiment would be more convincing.
- This paper uses a regression head to predict flow. However, many previous methods have shown the benefits of performing matching (for example, MASt3R, UniMatch, GMFlow, RoMa, etc.). It's unclear why such a design is adopted in this paper, and it would be better to justify the rationality and benefits of using a regression head instead of a matching head.
- Although the results on the wide-baseline feature matching task look good, the performance on optical flow seems to have room for further improvement (in particular, Table 4, Sintel Clean). What are the potential reasons? A deep analysis would be beneficial. In addition, how does the model predict flow for all pixels since the models is only trained on covisible pixels? In addition, I am also curious about the performance of the proposed method on the closed test sets (for example, Sintel and KITTI online benchmarks). It would be great if the authors could report these results.

---

> ### Author Rebuttal · Authors · 2025-07-31
>
> Thank you for your thoughtful and constructive comments. We are encouraged that you find the **unified model appealing**, and it **achieves comparable or better results than previous specialized methods**. We also appreciate your recognition of our **detailed data processing** as a **valuable contribution** to the community.
>
> We acknowledge that our writing has caused confusion regarding the term “coarse-to-fine” and address your concern that some architectural claims lack sufficient support from a more rigorously controlled experiment. We also investigated two reasons for your question on further improving flow performance, supported by evidence of improvement.
>
> - **Q1: Justification for UFM’s architecture**
>     - **Q1.1: Many prior methods demonstrate the advantages of using a matching head. The choice of a regression head in this work lacks clear justification.**
>
>         Matching can be performed either at the pixel level or the patch level.
>
>         **We do leverage pixel-level matching for precision.** As noted in lines 153–156, while regression offers robustness, it lacks precision. Therefore, we refine the regressed flow via local, pixel-wise, feature matching. Unlike MASt3R, which performs global matching (O(H^2W^2)), we restrict matching to a local window (O(HWP^2), P=7) initialized by the regressed flow, significantly improving efficiency while maintaining competitive accuracy.
>
>
>         **We chose not to perform patch-level matching based on empirical findings; as shown in Figure 5, UFM’s architecture better fits our combined training set compared to UniMatch and RoMa.**
>
>        To further support our argument, we observe a **22% increase in covisible EPE** when replacing the **regression DPT head of UFM** (patch level features decoded to pixel level fine flow) with the **coarse-to-fine matching head of UniMatch** (patch level coarse flow & features decoded to pixel level fine flow) when trained with the same setup of UFM560:
>
>         EPE [covis]|Sintel-C|Sintel-F|KITTI|ETH3D|DTU|TA-WB|
>         |-|-|-|-|-|-|-|
>         Regression - UFM|**0.79**|**1.44**|**1.87**|**2.64**|5.56|**12.87**|
>         Matching - UniMatch|0.91|1.49|2.05|5.36|**5.47**|13.85|
>
>     - **Q1.2: This paper argues about the difference with previous coarse-to-fine methods in several places, but eventually the proposed model still has a coarse-to-fine structure (with an additional refinement stage). I think this paper would like to highlight the difference in the “coarse” stage (global vs. local), instead of the overall coarse-to-fine structure in previous methods.**
>
>         We may differ in our definition of “coarse-to-fine,” and we admit the paper could define it more clearly. “Coarse-to-fine” can imply two things: (1) the predicted flow is upsampled from a coarse to a fine resolution, or (2) a rough prediction is progressively refined for higher precision. We think you are referring to (2). **While prior methods often combine both, our approach uses (2) but not (1).** We will clarify with the above description in the revision.
>
>     - **Q1.3: The argument on global attentions used in the model is not well-supported by experiments. In Figure 4, some comparisons with RoMa are shown, but there are many differences in implementation and training datasets between the proposed method and RoMa, which makes it not easy to understand where the performance gains come from.**
>
>         This is a fair point. We agree that the comparison with RoMa in Figure 4 does not conclusively isolate the effects of its coarse-to-fine strategy (definition (1) in Q1.2) due to differences in implementation and training setup.
>
>         **To better support our claim, we refer to the additional ablation study described in Q1.1, where we directly compare regression to a coarse-to-fine head under the same training conditions.** We acknowledge that the original justification was primarily based on visual intuition and theoretical considerations of the RoMa architecture. While this motivation remains valid, the new empirical results in Q1.1 now offer stronger support for our design choice. **We will revise our original justification with this evidence to better support it.**
>
> - **Q2: Analysis for further improving Optical Flow performance (especially, Sintel Clean).**
>
>     We observed two directions for improvement:
>
>     - **We show that finetuning the backbone during refinement training improves the precision of small motions, yielding a 10% gain on Sintel Clean.** Previously, we trained UFM980-Refine in two stages: first, training the flow head, then freezing the backbone to train the refinement. We now train both jointly for 15 epochs (matching the first stage of the original scheme) and report the updated results in Table 4:
>
>         | |Sintel-Clean| | |Sintel-Final| | |KITTI| | |
>         |-|-|-|-|-|-|-|-|-|-|
>         | | EPE[covis] | EPE[all] | 1/3/5 px | EPE[covis] | EPE[all] | 1/3/5 px |EPE[covis]|EPE[all]|F1 %|
>         UFM_980-Refine|0.56|1.15|10.2/4.6/3.3|**1.25**|2.01|15.0/7.2/**5.1**|**2.05**|2.96|11|
>         UFM-980-Refine[backbone FT]|**0.5** |**1.02**|**9.1/4.1/2.8**|**1.25**|**1.98**|**14.0/7.0/5.1**|2.176|**2.9**|**10.366**|
>
>         As shown above, this joint training yields improvement on most EPE and inliers, with the most improvement on easier datasets like Sintel Clean. Hence, we conclude that training the refinement stage jointly with the backbone enables the model to learn more detailed matching features.
>
>     - Our method is **biased towards static objects**. According to KITTI performance in Q2.1, UFM is significantly better in the background than in the foreground objects, which are mainly dynamic. We believe this is a data issue, as less than 20% of our training data is dynamic. Future work may include more sceneflow datasets, such as ParallelDomain, to balance the distribution. We will update the limitations to include this analysis.
>
>
>     - **Q2.1: Performance in Sintel / Kitti test sets.**
>
>         **UFM outperforms UniMatch on Sintel and KITTI test sets under zero-shot settings.** We benchmarked the UFM-980 refine model and UniMatch reported in Table 4 on the Sintel and KITTI benchmark test sets. **Neither method was trained on Sintel or KITTI**.
>
>         **Sintel Test set result**:
>
>         ||EPE all|EPE Matched|EPE unmatched|d0-10|d10-60|d60-140|s0-10|s10-40|s40+|
>         |-|-|-|-|-|-|-|-|-|-|
>         |Sintel-Final||||||||||
>         |UFM_980-Refine|**3.282**|**1.746**|**15.802**|**3.28**|**1.273**|**1.205**|**0.689**|**2.068**|**19.067**|
>         |UniMatch-CT|4.139|1.965|21.856|3.765|1.586|1.278|0.888|2.426|24.401|
>         |Sintel-Clean||||||||||
>         |UFM_980-Refine|**1.682**|0.718|**9.518**|1.535|0.666|0.425|0.483|1.231|**8.705**|
>         |UniMatch-CT|1.802|**0.571**|11.845|**1.175**|**0.453**|**0.395**|**0.367**|**0.932**|11.032|
>
>         From the Sintel result, UFM outperforms UniMatch in the overall metric and most sub-items, except for sub-items on Sintel Clean, which we demonstrated in Q2 can be improved through training the refinement in parallel with the backbone.
>
>         **KITTI Test set result**
>
>         ||Range|F1-bg|F1-fg|F1-all|
>         |-|-|-|-|-|
>         UFM_980-Refine|All|**8.67**|22.08|**10.9**|
>         UniMatch-CT|All|18.01|**17.27**|17.89
>         UFM_980-Refine|Not occluded|**4.93**|19.6|**7.59**
>         UniMatch-CT|Not occluded|9.75|**14.61**|10.63
>
>         From the KITTI result, UFM outperforms UniMatch in the overall metric and background objects, but not foreground objects, which are mostly dynamic. As discussed in Q2, we believe adding more dynamic data will improve its performance.
>
> - **Q3: Discussing PerceiverIO**
>
>     Thank you for the reference - we will cite it in the discussion on scaling correspondence.
>
> - **Q4: Typo**
>
>     Thank you for spotting this. We will correct it.

---

> > ### Comment · Reviewer_RKdL · 2025-08-04
> >
> > I thank the authors for the rebuttal. I am still trying to understand the experiments presented in Q1.1, where the authors "replaced the regression DPT head of UFM (patch level features decoded to pixel level fine flow) with the coarse-to-fine matching head of UniMatch (patch level coarse flow & features decoded to pixel level fine flow)". Since I think the comparison between matching and regression is important to gain insights on these two approaches, I would like to double check the experimental setup:
> >
> > - I assume the two experiments are conducted with the final flow prediction head as the only difference?
> > - What does "coarse-to-fine matching head of UniMatch" mean exactly? How is the coarse flow estimated and how is the refinement preformed?
> >
> > In addition, I feel at least matching would be superior in handling large motions, as evidenced by the success of wide-baseline feature matching methods like MASt3R and RoMa. How's the authors' opinion on this? Thanks.

---

> > > ### Author Response · Authors · 2025-08-04
> > >
> > > Thank you for your additional comments. We provide further clarification on the coarse-to-fine ablation settings below, along with additional insights.
> > >
> > > **Q5.1: I assume the two experiments are conducted with the final flow prediction head as the only difference? What does "coarse-to-fine matching head of UniMatch" mean exactly? How is the coarse flow estimated, and how is the refinement performed?**
> > >
> > > **Yes, the dataset, training scheme, and the model architecture until the global attention transformer outputs are the same, as training UFM560 described in the paper & Figure 2. The final flow prediction head & how the flow loss is applied are the only differences.** In particular, we also modified the UFM robust loss to match how UniMatch applies the loss. The key is we supervise on two outputs as UniMatch: bilinearly upsampled coarse flow ($\hat \phi_{patch}$ below) from patch-wise feature matching and the predicted upsampled fine flow after refinement, $\phi$.
> > >
> > >
> > > We define the UFM head and UniMatch head as follows:
> > >
> > > - **UFM head:** Given transformer features $F_1 \in \mathbb R^{\frac{H}{P} \times \frac{W}{P}\times 768}$ for Image 1, a DPT network $f_{DPT}$ directly regress it to pixel-wise flow.
> > >
> > >     $$\phi_{UFM} = f_{DPT}(F_1)$$
> > >
> > > - **UniMatch head:** **We used the original code from UniMatch for its head.** We present a concise summary below (for further info please refer to formulas (1) - (7) from GMFlow for steps 1, 2, and Section 3.3, plus Fig. 8 from RAFT for step 3):
> > >     1. **Compute coarse flow from patch-wise feature similarity.**
> > >         Given info-sharing features from the source and the target branch $F_1, F_2 \in \mathbb R^{\frac{H}{P} \times \frac{W}{P}\times 768}$, and patch coordinates $G \in \mathbb R^{\frac{H}{P} \times \frac{W}{P}\times 2}$ we compute the coarse flow from feature similarity as:
> > >
> > >         $$\hat \phi_{patch} = \text{softmax}(F_1 F_2^T)\cdot G - G$$
> > >
> > >         In other words, for each source patch, we compute its dot-product similarity to all the patch features in the target branch, take the softmax, and weight-add the target coordinate. We then subtract the source patch's coordinate to obtain its coarse flow.
> > >
> > >         **This step is representative of matching in the patch scale.**
> > >     2. **Feature-Flow attention.**
> > >         This step is unique to UniMatch. It first creates projected features $F_{1, Q}, F_{1, K}$ from 2 linear projections of $F_1$, and then computes
> > >
> > >         $$\phi_{patch} = \text{softmax}(F_{1, Q}F_{1, K}^T) \times \hat \phi_{patch}$$
> > >
> > >         The motivation is to propagate the coarse flow for occluded regions based on structural similarity in the source image. We refer to Section 3.3 of GMFlow for further clarification.
> > >
> > >     3. **Convex upsampling to pixel-wise flow.**
> > >         The source feature $F_1$ is regressed to weights $\hat W\in\mathbb R^{H \times W \times 3 \times 3}$ and normalized with softmax on its last two dimensions, creating 3x3 weights that sum to 1. We denote it by $W$. The pixel-wise flow at $(i, j)$ is computed as
> > >         $$\phi_{i,j} = \sum_{a=-1}^1 \sum_{b=-1}^1 P \cdot \phi_{patch}[p_i + a, p_j + b] \cdot W[i, j, a, b]$$
> > >
> > >         Where $(p_i,p_j) = (i//P,j//P)$ is the patch index of pixel $(i, j)$. In other words, the linear projection creates normalized  $3\times 3$ weights that determine how each pixel combines the $3 \times 3$ coarse flows from adjacent patches into its own flow. The $P$ multiplier is used to convert patch coordinates into equivalent pixel coordinates.
> > >
> > > Essentially, as a combination of these 3 steps, UniMatch head first performs patch-level (coarse) matching which is then upsampled to fine correspondence/flow. The architecture supervises both the patch-level matching and pixel-level upsampling, which is similar in motivation to other matching based architectures like RoMa.
> > >
> > > As shown in the previous rebuttal response, we found this head to be empirically suboptimal than directly regressing pixel-level flow from the transformer features.

---

> > > > ### Author Response · Authors · 2025-08-04
> > > >
> > > > **Q5.2: In addition, I feel at least matching would be superior in handling large motions, as evidenced by the success of wide-baseline feature matching methods like MASt3R and RoMa. How's the authors' opinion on this? Thanks.**
> > > >
> > > > In general, we believe Table 2 and the rebuttal's coarse-to-fine ablation experiment already demonstrate that UFM has better performance even at handling large motions, as the benchmarking datasets DTU, ETH3D, and TA-WB have mean, std, and max pixel flows of (85.7+-70.1, max=620.9), (157.7+-115.6, max=622), and (192.28+-99.94, max=669.69), which is very large motion on average.
> > > >
> > > > Our opinion as to why we think the regression paradigm in UFM is better is as follows:
> > > >
> > > > - In comparison to RoMa or other coarse-to-fine architectures in general (patch-wise matching + upsample), the clarification in Q5.1 above shows that our ablated UniMatch head is representative of this architecture and that direct regression is empirically better. We also find a similar trend in Figure 5 of the paper using RoMa with our data.
> > > >
> > > > - With regards to MASt3R (pixel-wise global matching vs. regression), we acknowledge that the overall architecture is very similar. However, we believe the core difference lies in **supervision efficiency**. Since MASt3R relies on global matching, requiring computing feature similarity with all pixels during supervision, MASt3R selects at most 4096 pairs of correspondence per image, whereas UFM leverages loss on all (covisible) pixels (much larger than 4096).
> > > >
> > > >     Empirically, as supported by Table 2, UFM is comparable to MASt3R with a significant advantage in speed and completeness of the prediction in dense applications. From a simplicity and scalability perspective, we believe that the regression design choice of UFM is preferable.
> > > >
> > > > **On top of the above discussion, we also show that matching and regression are not mutually exclusive.** Matching-based local refinement can be combined with the regressed fine flow (another key difference of UFM-Refine with prior dense correspondence methods) to improve precision further.

---

> > > > > ### Comment · Reviewer_RKdL · 2025-08-04
> > > > > **Thanks for the clarifications**
> > > > >
> > > > > I thank the authors for the clarifications and my questions are well-addressed. Therefore, I am happy to raise my score. I think this paper would become stronger by including these clarifications and discussions in the revised version.

---

### Official Review · Reviewer_vKeR · 2025-06-30

**Clarity:** 4
**Significance:** 4
**Originality:** 3
**Rating:** 4
**Confidence:** 5

**Summary:**

Similar to UniMatch and MASt3R, this paper builds a dense correspondence network based on Dinov2 and Transformer. Additionally, it introduces a local attention-based refinement module to further enhance the results. Overall, the network architecture is reasonable, but it lacks substantial innovation. However, I believe the more significant contribution of this paper lies in the extensive training over a wide range of datasets, which provides a potentially reliable matching baseline for many downstream tasks.

**Questions:**

1. Please elaborate on the algorithm’s performance in occluded regions, and explain why occlusion estimation is not considered.

2. Please describe your open-sourcing plan. Since the main contribution of this work lies in its applicability to downstream tasks, a well-executed open-source release would be a major positive factor.

**Ethical Concerns:**

["NO or VERY MINOR ethics concerns only"]

**Limitations:**

yes

**Quality:**

4

**Strengths And Weaknesses:**

**Strengths**
1. Once the code is open-sourced, it will provide an excellent matching tool for many downstream tasks, which is beneficial to the community.

2. The overall network structure and the proposed refinement module are reasonable.

3. Extensive experiments and thorough visualizations are provided, and the results appear trustworthy.

**Weaknesses**
1. The method overlooks the learning of occluded regions. Inferring occlusions is a crucial aspect in optical flow tasks, especially for downstream tasks such as sparse-view reconstruction. This omission reduces the significance of the work (as its functionality overlaps with MASt3R in most practical scenarios).

2. The architecture lacks substantial innovation. Technically, the method appears to be a combination of UniMatch and MASt3R.

3. The paper’s claim of unifying dense correspondence and flow is not well justified. At most, it improves long-range and short-range matching performance in dense correspondence tasks. Optical flow generally includes occlusion estimation, which is not addressed here.

---

> ### Author Rebuttal · Authors · 2025-07-31
>
> Thank you for your thoughtful and detailed comments. We are heartened by your recognition of the model's **wide applicability to downstream tasks**, coming from the contribution of **extensive training over a wide range of datasets**. We appreciate your validation that **the results are trustworthy** based on the **extensive experiments and thorough visualizations**.
>
> We have addressed your questions/clarifications on performance in occluded regions, our unification claim, lack of architectural innovation, and open source plan.
>
>
> - **Q1: Questions regarding performance in occluded regions.**
>     - **Q1.1: The method overlooks learning in occluded regions, which is a crucial aspect in optical flow tasks.**
>
>
>         **We believe that this might not have been very clear in the paper, and would like to clarify that we have trained and benchmarked optical flow performance in all pixels (including occluded) and showed SoTA performance.** We recognize that optical flow requires estimating occluded and out-of-view pixels. Unlike the 560-resolution model, **we trained the 980-resolution model on all pixels**, as noted in **lines 225-226**. **We evaluated it on all pixels** following standard protocol, as shown in **Table 4** (**“EPE[all]” and “1/3/5 px”** entries). The 980 model achieves state-of-the-art zero-shot performance on Sintel-Final and KITTI, with competitive results on Sintel-Clean (lines 260-262). Notably, it was bootstrapped from the 560 model using only 15 epochs of finetuning, demonstrating UFM's adaptability to occlusions for optical flow.
>     - **Q1.2: The paper's claim of unifying dense correspondence and flow is not well justified, because optical flow generally includes occlusion estimation, which is not addressed here.**
>
>         While we evaluated UFM concurrently on covisible regions of optical flow and wide-baseline datasets and demonstrated SoTA, we agree that optical flow tasks, in the strict sense, include occluded and out-of-view pixels. We address this limitation by simply finetuning our model, as noted in our response to Q1.1.
>
>
>         **We show our model can be easily finetuned to maintain our unification claim.** We finetuned the UFM560 checkpoint for 10 epochs and supervised on all pixels for the optical flow datasets and covisible pixels in the wide-baseline dataset. Specifically, we used the same number of pairs of optical flow data and 20% of wide-baseline data as specified in Table 1, and the same hyperparameter with a 10x smaller learning rate as training the 560 checkpoint (Section 3.4). We repeat benchmarks in Table 2 and 4:
>
>
>
>         **Repeated Table 2** (We omit OF baselines due to significantly worse performance)
>
>         |Method|Eval Range|ETH3D EPE|1/2/5 px|DTU EPE|1/2/5 px|TA-WB EPE|1/2/5 px|Runtime|
>         |-|-|-|-|-|-|-|-|-|
>         RoMa|covisible|7.94|51/33/20|9.69|52/34/20|48|64/48/40|387.4|
>         UFM_560|covisible|2.64|47/24/9|5.56|58/34/13|12.87|54/32/17|42.9|
>         UFM_560-Refine |covisible|2.60|44/23/9|5.55|56/33/14|12.84|51/31/17|70.1|
>         UFM_560-FT|covisible|2.917|47/25/10|5.936|59/34/14|13.65|54/32/17|42.9|
>         MASt3R|MASt3R’s selection|1.31|33/12/**2**|2.23| 50/**21/5**|6.21| 55/23/**6**|2517.8|
>         UFM_560|MASt3R sel|1.34|32/12/3|2.30|49/24/6|6.19|42/20/7|41.0|
>         UFM_560-Refine|MASt3R sel|**1.29**|**29/11**/3|**2.18**|**43/21**/6|**6.13**|**39/18**/7|56.1|
>         UFM_560-FT|MASt3R sel|1.422|31/13/4|2.418|50/24/7|6.394|42/20/7|41.6|
>
>         **Repeated Table 4**:
>
>         |Method | Sintel-C EPE[covis] | S-C EPE[all] | S-C 1/3/5 px | S-F EPE[covis] | S-F EPE[all] | S-F 1/3/5 px | KITTI F1 EPE[covis] | KITTI F1 EPE[all] | KITTI F1 % | Runtime |
>         |-|-|-|-|-|-|-|-|-|-|-|
>         SEA-RAFT | 0.65 | 1.47 | 11/5/**3** | 2.24 | 3.69 | **16**/9/7 | 2.36 | 4.21 | 15.5 | 14.7|
>         FlowFormer | 1.88 | 2.92 | 24/10/7 | 7.39 | 8.92 | 35/22/18 | 4.64 | 7.89 | 29.3 | 77.5 |
>         UniMatch | **0.60** | **1.2** | **10/4/3** | 1.73 | 2.76 | **16/8/6** | 2.43 | 4.66 | 17.7 | 30.0 |
>         RoMa|1.18|(2.913)|(16/9/7)|2.13|(3.99)|(25/14/11)|2.30|(10.98) | (19.6) | 390.3 |
>         UFM_560 | 0.79 | (2.04) | (17/7/4) | 1.44 | (2.93) | (20/10/7) | 1.87 | (9.15) | (19.0) | 44.0 |
>         UFM_560-Refine | 0.72 | (1.98) | (14/6/5) | **1.40** | (2.90) | (19/9/7) | **1.69** | (9.02) | (17.51) | 57.0 |
>         UFM_560-FT | 0.78 | 1.358 | 16/6/4 | 1.47 | **2.28** | 20/9/**6** | 1.96 | **4.14** | **14.8** | 45.9 |
>
>         Where result in () denotes the method is not trained in occluded and out-of-view regions.
>
>
>         **For dense wide-baseline, UFM560_FT still achieves 58% less EPE and 8.5x less runtime compared to the closest performing dense method, RoMa.** Despite evaluating on the pixels that MASt3R is confident about (it uses a cycle consistency check to filter its outputs, explained in lines 236-239), UFM560_FT still achieves comparable metrics while being 54x faster.
>
>
>         **For optical flow, UFM achieves better covisible and all EPE and similar outlier rates in Sintel-Final and KITTI, and comparable performance in Sintel-Clean.**
>         Note that our **refinement module can improve this result** further. Thus, the above experiment shows UFM can be easily finetuned to concurrently solve wide-baseline matching and optical flow in the strict sense.
>
>     - **Q1.3: The omission of learning in occluded regions reduces the significance of the work (as its functionality overlaps with MASt3R in most practical scenarios).** Explain why occlusion estimation is not considered. Please elaborate on the algorithm's performance in occluded regions.
>         **The question remains for wide-baseline** since we addressed it for optical flow in Q1.1 and Q1.2.
>
>
>         Certainly, being able to predict correspondence in occluded regions broadens UFM's applications. However, estimating occluded regions is hard in wide-baseline datasets, and **it is not the norm for dense-wide baseline matching methods (e.g.,  DKM, RoMA, Dgc-net) to estimate occluded regions (as lines 89-91)**. We find that including occluded regions will decrease performance in covisible regions, as partially shown in Q1.2. **Still, including occluded regions would be an interesting future direction for UFM.**
>
>         **We benchmark UFM and RoMa (the closest SoTA dense matcher) on occluded regions and show UFM still outperforms RoMa.** Note that neither method has been trained on occluded regions.
>
>
>         EPE | ETH3D | | DTU | | TA-WB | |
>         |-|-|-|-|-|-|-|
>         Range | covisible | occluded | covis | occ | covis | occ |
>         RoMa | 7.94 | 8.67 | 9.69 | 49.72 | 48.10 | 86.53 |
>         UFM-560 | 2.64 | 3.926 | 5.56 | **25.64** | 12.87 | **55.45** |
>         UFM-560 Refine | **2.60** | **3.86** | **5.55** | **25.64** | **12.84** | 55.52 |
>
> - **Q2: The architecture lacks substantial innovation. Technically, the method appears to be a combination of UniMatch and MASt3R.**
>
>     We respectfully argue that UFM is **not merely a combination of UniMatch and MASt3R**, as it **distills only their most relevant ideas** and, more importantly, **introduces its innovations aimed at achieving simplicity, efficiency, and scalability**. We do acknowledge that UFM draws inspiration from prior architectures in lines 156 - 162.
>
>     **Base Module**: As shown in the architecture ablation in **Figure 5**, UFM outperforms representative architectures (including UniMatch) when trained on the same unified dataset, demonstrating better scalability.
>
>     **Compared to MASt3R, UFM introduces key abilities not achievable by MASt3R’s architecture:**
>      - **1. Efficiency**: MASt3R performs global pixel-wise matching (complexity O(H^2W^2)). UFM avoids this by initializing with regressed pixel-level flow (in contrast to patch level for UniMatch) and performing optional local refinement (complexity O(HWP^2), with P=7), enabling efficient and high-res prediction.
>      - **2. Completeness**: MASt3R enforces one-to-one matching, which results in incomplete dense predictions or slower inference without its fast reciprocal matcher. UFM makes no such tradeoff.
>      - **3. Sub-pixel Matching**: MASt3R's proposed matcher predicts integer coordinates only, lacking potential for sub-pixel prediction (which is in contrast to UFM, where this is also architecturally possible with our refinement as discussed in Appendix D).
>
>     Additionally, UFM simplifies MASt3R's transformer: we remove RoPE (which we found not required), use global attention (for better acceleration from Flash Attention, as explained in lines 142-145), and use DINOv2 to ensure better semantic matching ability over MASt3R (please refer to our answer to Q1 of R1(B4yj)).
>
>
>     **Compared to UniMatch**, UFM does not construct its flow from coarse-to-fine resolutions, **which we found suboptimal** for the unified training (Figure 5). To further support our argument, we observe a **22% increase in covisible EPE** when replacing the **regression DPT head of UFM** (patch level features decoded to pixel level fine flow) with the **coarse-to-fine matching head of UniMatch** (patch level coarse flow & features decoded to pixel level fine flow) when trained with the same setup of UFM560:
>
>     EPE [covis]|Sintel-C|Sintel-F|KITTI|ETH3D|DTU|TA-WB|
>     |-|-|-|-|-|-|-|
>     Regression - UFM|**0.79**|**1.44**|**1.87**|**2.64**|5.56|**12.87**|
>     Matching - UniMatch|0.91|1.49|2.05|5.36|**5.47**|13.85|
>
> - **Q3: Please describe your open-sourcing plan.**
>
>     As stated in Checklist #5 (**line 540 - 552**), we will be releasing our training and inference code, dataset preparation and experimental evaluation scripts, pretrained model weights, as well as usage instructions after acceptance. **Practically, we plan to release model weights and inference code immediately after the review process and the rest within the camera-ready deadline.**

---

> > ### Comment · Reviewer_vKeR · 2025-08-07
> >
> > The authors’ rebuttal has addressed some of my concerns. Although the level of novelty is limited, the work still provides value to the community. I will maintain my weak accept rating.

---

### Official Review · Reviewer_B4yj · 2025-07-02

**Clarity:** 4
**Significance:** 4
**Originality:** 3
**Rating:** 5
**Confidence:** 3

**Summary:**

This paper proposes utilizing both optical flow and large baseline correspondence datasets for mutual benefits on two tasks.
The authors present a simple but effective model architecture which consists of visual encoder, 12-layer transformer (with the positional encoded features of the two encoders as input) and two heads for flow and co-visibility (with encoder output and 6/9/12 layer outputs as input). The output of regression flow head is further fine-tuned via a weighted summation of neighbor coordinates based on feature similarity between reference and target frames.
In training, only co-visible pixels are utilized. In order to identify co-visible pixels, according to the supplementary, depth reprojection error is basically used and, additionally, information on depth change (in dynamic scenes) and rigidity (in rigid motion) is also utilized.
By applying the proposed geometric sampler, the authors curated the existing datasets to obtain a new selected set (TA-WB) of data to evaluate dense wide-baseline correspondence solutions.

**Questions:**

1) How can we further investigate the benefits of multi-task learning in this paper?

2) How does the proposed method work in extreme conditions such as low light scenes?

**Ethical Concerns:**

["NO or VERY MINOR ethics concerns only"]

**Final Justification:**

The authors well addressed my concerns in the rebuttal. Thus, I want to keep the score.

**Limitations:**

Yes

**Quality:**

3

**Strengths And Weaknesses:**

Strength

1) Simple but effective method
Multi-task benefits are well known. But, the authors demonstrated how to realize the benefits in optical flow and large baseline correspondence with an effective method.

2) Experiments are well done.
Especially, probing experiments well demonstrate the authors' choice in model architecture.

Weakness

1) Lack of detailed analysis on the origin of multi-task learning effects
Table 5 well demonstrated the mutual benefits.
It would be nice to further investigate the origin of such benefits.

2) Evaluation of diverse scenes
Although the authors proposed a new dataset for wide-baseline correspondence,
it would be nice if more diverse scenes like low light, fast moving objects, etc., are evaluated.

---

> ### Author Rebuttal · Authors · 2025-07-31
>
> Thank you for your thoughtful feedback and for taking the time to read the supplementary material in detail. We are encouraged that you find our method to be **simple, effective, and achieves mutual benefits** by combining optical flow and wide-baseline matching tasks and data. We are also thrilled that you find our **experiments well done** and that the **model architecture design is well supported**, especially by the probing experiment.
>
> We have added results to address your question about performance in extreme cases and your suggestions for further improvement:
>
> - **Q1: What is the model’s performance under extreme conditions, such as low-light scenes?**
>
>     - **We add the WxBS Benchmark and show that UFM is reasonably robust to extreme condition shifts.** The benchmark includes low-light scenes, multiple visual spectrums (e.g., IR to RGB), day-night transitions, seasonal changes, and scale changes. We report performance on fundamental matrix estimation using PCK, i.e., the percentage of pre-labeled keypoints consistent with the prediction under an epipolar error threshold (higher is better, 10px is commonly used).
>
>         | Thresholds | 3px  | 5px  | 10px | 15px | 19px |
>         |------------|------|------|------|------|------|
>         | RootSIFT   | 0.06 | 0.08 | 0.10 | 0.13 | 0.16 |
>         | Superpoint + SuperGlue | 0.26 | 0.33 | 0.46 | 0.51 | 0.55 |
>         | LoFTR      | 0.48 | 0.57 | 0.67 | 0.71 | 0.73 |
>         | RoMa       | **0.74** | **0.85** | **0.92** | **0.96** | **0.96**|
>         | MASt3R     | 0.32 | 0.42 | 0.56 | 0.62 | 0.65 |
>         | UFM_560    | 0.49 | 0.63 | 0.74 | 0.77 | 0.78 |
>         | UFM_560-Refine | 0.49 | 0.62|  0.75 | 0.79 | 0.81 |
>
>         As already noted in Section 5 (Limitations), compared to RoMA (the best performing one, where DINOv2 is frozen), finetuning the DINOv2 encoder in UFM improves precision but reduces semantic matching capabilities (as also supported by our supplementary material analysis, Section E). Despite this, UFM ranks second on the WxBS benchmark with a 0.74 PCK@10, showing strong performance and outperforming MASt3R under extreme conditions.
>     - **UFM also has reasonable performance for low-light conditions**, as UFM succeeded (matched >50% keypoints with a 10 px error) in 3/5 IR-RGB matches (note that there are no IR images in our training) and 9/11 day to night matches.
>     - **We had some early exploration for further improvement.** For low-light conditions, we tried adding a dark augmentation (p = 0.005) to the training pipeline, which simulates extreme dark and quantization effects, and observed improved IR matching (4/5) in WxBS. More fundamentally, we believe that this is related to encoder finetuning, and we think future work can explore semantic preservation losses, such as those used in Depth-Anything V2 and DUNE.
>     **We will add the WxBS result (also includes qualitative visualizations) and expand the limitations section accordingly.**
>
> - **Q2.1: Lacking analysis of the source of mutual benefit** when training for optical flow and wide-baseline dense correspondence (shown in Table 5).
>
>     **While not a direct analysis, our architecture ablation (Figure 5) indicates that UFM’s architecture is the most effective in scaling to the unified dataset**, which we believe is crucial for achieving mutual benefits across optical flow and wide-baseline matching. We will state this explicitly in the revision.
>
> - **Q2.2: Also, how can we further investigate the benefits of multi-task learning?**
>
>     While not a focus of this paper, we believe the model may benefit further from multi-task training, such as combining 3D or 4D tasks with correspondence. We are interested in investigating this in future work.
>
> - **Q3: It would be nice if more diverse scenes, like low light, fast-moving objects, etc., were evaluated.**
>
>     - **We agree, and we will incorporate the above WxBS results into the revision.**
>     - **The TA-WB benchmark lacks moving objects** because the source data lacks sceneflow, preventing accurate covisibility computation. Still, it is a **geometrically challenging** benchmark as visualized in Supplementary Figure S.2. We will expand the limitations section to include this point about our benchmark and we believe that future work can explore converting dynamic datasets or simulators to generate more challenging pairs using our geometric sampler.

---

> > ### Comment · Reviewer_B4yj · 2025-08-05
> >
> > Thank you for the rebuttal. It addressed major concerns of mine and I hope the authors will incorporate them into the revised manuscript.

---

### Decision · Program_Chairs · 2025-09-17

**Decision:**

Accept (poster)

**Comment:**

The paper introduces a dense correspondence network based on Dinov2 and Transformer. In particular, it has a local attention-based refinement module to further enhance the results. Although the contributions on the network architecture is not significant, the network offers superior performance by extensive training over a wide range of datasets. The resulting baseline can benefit many downstream tasks. The reviewers agreed to accept after the rebuttal session. The AC agrees with the decision.